# UGMathBench: A Diverse and Dynamic Benchmark for Undergraduate-Level Mathematical Reasoning with Large Language Models

Xin Xu*, Jiaxin Zhang*, Tianhao Chen*, Zitong Chao, Jishan Hu, Can Yang

Department of Mathematics
The Hong Kong University of Science and Technology
`{xxuca, jzhangat, tchenbb, zchaoaa}@connect.ust.hk`
`{majhu, macyang}@ust.hk`

## Abstract

Large Language Models (LLMs) have made significant strides in mathematical reasoning, underscoring the need for a comprehensive and fair evaluation of their capabilities. However, existing benchmarks often fall short, either lacking extensive coverage of undergraduate-level mathematical problems or probably suffering from test-set contamination. To address these issues, we introduce UGMathBench, a diverse and dynamic benchmark specifically designed for evaluating undergraduate-level mathematical reasoning with LLMs. UGMathBench comprises 5,062 problems across 16 subjects and 111 topics, featuring 10 distinct answer types. Each problem includes three randomized versions, with additional versions planned for release as leading open-source LLMs become saturated in UGMathBench. Furthermore, we propose two key metrics: effective accuracy (EAcc), which measures the percentage of correctly solved problems across all three versions, and reasoning gap ($\Delta$), which assesses reasoning robustness by calculating the difference between the average accuracy across all versions and EAcc. Our extensive evaluation of 23 leading LLMs reveals that the highest EAcc achieved is 56.3% by OpenAI-o1-mini, with large $\Delta$ values observed across different models. This highlights the need for future research aimed at developing "large reasoning models" with high EAcc and $\Delta = 0$. We anticipate that the release of UGMathBench, along with its detailed evaluation codes, will serve as a valuable resource to advance the development of LLMs in solving mathematical problems. Codes and data are available at https://github.com/YangLabHKUST/UGMathBench.

## 1 Introduction

Mathematical reasoning and problem-solving are critical components of human intelligence, and the ability of machines to understand and address mathematical challenges is crucial for their deployment (Ahn et al., 2024; Liu et al., 2024; He et al., 2024a). Solving mathematical problems with machines has been a significant research topic in natural language processing since the 1960s (Bobrow et al., 1964), initially focusing on elementary math word problems (Patel et al., 2021; Wang et al., 2017; Ling et al., 2017; Welbl et al., 2017; Cobbe et al., 2021). With the advent of Large Language Models (LLMs) (Brown et al., 2020; Ouyang et al., 2022; OpenAI, 2023; Team et al., 2023; Anthropic, 2024), interest in using these advanced technologies to solve math problems has continued to grow. Researchers are exploring various approaches to improve the mathematical reasoning capabilities of LLMs, including prompting (Wei et al., 2022; Wang et al., 2023b; Kojima et al., 2022; Zhang et al., 2023; Zheng et al., 2023; Xue et al., 2024), supervised fine-tuning (Yue et al., 2023; Yu et al., 2023; Gou et al., 2023; Li et al., 2024a; Xu et al., 2024b; Tong et al., 2024; Yan et al., 2024; Xu et al., 2025b; Yan et al., 2025), and continued pretraining (Lewkowycz et al., 2022; Shao et al., 2024;

---

* Equal contribution.

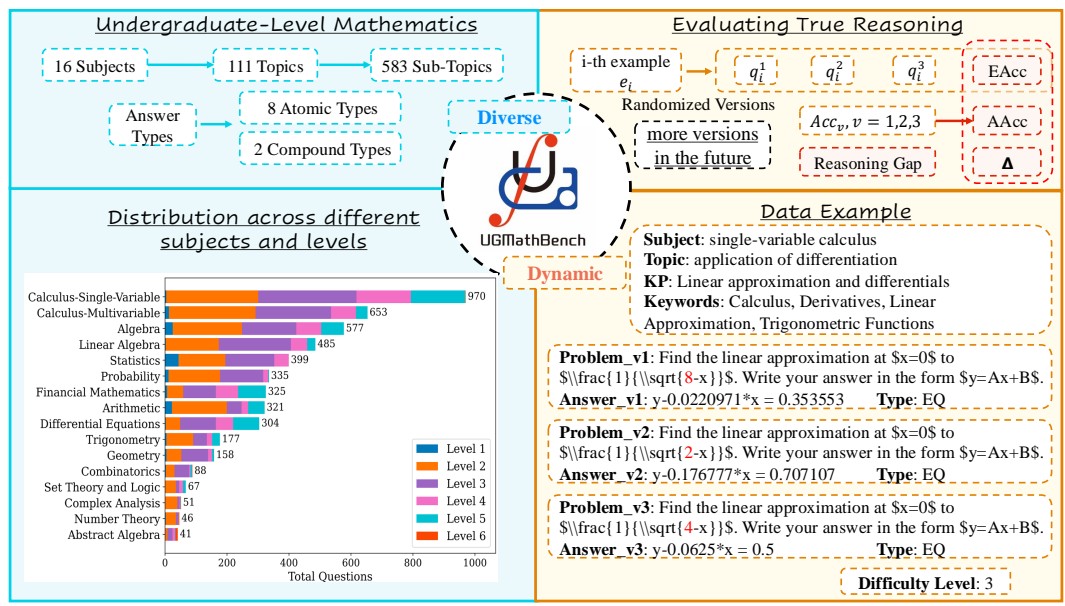

Figure 1: Overview of UGMathBench. UGMathBench is a diverse and dynamic benchmark specifically designed for evaluating undergraduate-level mathematics with LLMs, covering 16 distinct subjects and featuring 10 different answer types. Each problem contains three randomized versions, with EAcc and $\Delta$ rigorously assessing LLMs' true reasoning skills.

Azerbayev et al., 2023). Consequently, LLMs have become increasingly capable of solving complex mathematical problems (Hendrycks et al., 2021b; Ahn et al., 2024).

With the rapid advancements in LLMs, evaluating their reasoning capabilities has become increasingly important (He et al., 2024a; Xu et al., 2025a; Phan et al., 2025), especially in mathematics (Liu et al., 2024; Gao et al., 2024). Although benchmarks such as GSM8K (Cobbe et al., 2021) and MATH (Hendrycks et al., 2021b) are commonly used to assess these abilities, they are becoming insufficient due to rapid progress in model performance, as evidenced by accuracy exceeding 97% in GSM8K (Zhou et al., 2023) and 94.8% in MATH (OpenAI, 2024b). While more challenging benchmarks are being introduced (Tang et al., 2024; Liu et al., 2024), they often remain limited in size and scope regarding undergraduate-level mathematics, which is essential due to its breadth and complexity (see Table 1). Moreover, there are growing concerns about test set contamination in these static benchmarks (Srivastava et al., 2024; Zhang et al., 2024; Qian et al., 2024; White et al., 2024). Recent efforts Qian et al. (2024); Srivastava et al. (2024) have introduced dynamic benchmarks by functionalizing the original problems in GSM8K and MATH to generate randomized variations through variable disturbance. However, these initiatives primarily focus on elementary and competition-level mathematics (see Table 1). These limitations highlight the pressing need for a comprehensive and dynamic benchmark specifically designed to assess undergraduate-level mathematical reasoning.

In this paper, we present UGMathBench, a diverse and dynamic benchmark designed to evaluate the mathematical reasoning capabilities of LLMs across a wide range of undergraduate-level mathematical topics, as illustrated in Figure 1. We meticulously collect, clean, and format undergraduate-level mathematical problems from our online homework grading system (see Appendix B.1), resulting in a benchmark comprising 5,062 problems in 16 subjects, categorized into eight atomic answer types and two compound answer types. A key feature of UGMathBench is the inclusion of multiple randomized versions for each problem, which aids in assessing the true reasoning abilities of LLMs through the **E**ffective **Acc**uracy (EAcc) and reasoning gap ($\Delta$) (see Section 3.3). EAcc represents the percentage of problems correctly solved across all versions, providing insights into intrinsic reasoning skills. It operates on the premise that a model capable of solving a problem through reasoning should also be able to solve all its variants under variable disturbance (Srivastava et al., 2024; Qian et al., 2024). The reasoning gap, $\Delta$, is defined as the difference between the average accuracy across all versions and the EAcc, quantifying the robustness of reasoning when the original problems undergo slight

modifications. These metrics help mitigate the impact of potential test set contamination (Deng et al., 2024; Dong et al., 2024; Golchin & Surdeanu, 2023; Roberts et al., 2023) and ensure a more rigorous evaluation of LLMs' mathematical reasoning abilities. These features are summarized more clearly in Figure 1 and Table 1.

We conducted an extensive evaluation of the leading LLMs, including proprietary models such as OpenAI-o1 (OpenAI, 2024b) and open-source models like LLaMA-3-Instruct (AI@Meta, 2024). Despite their advanced capabilities, the best EAcc achieved is 56.3% by OpenAI-o1 (OpenAI, 2024b) and all LLMs exhibit a large reasoning gap. These results highlight the considerable challenges that UGMathBench presents to current LLms in terms of mathematical reasoning, underscoring the need for future research focused on developing "large reasoning models" characterized by high EAcc and a reasoning gap $\Delta = 0$. To summarize our key findings:

1. Even LLMs with the most advanced reasoning ability, OpenAI-o1-mini, achieves a 56.30% EAcc on UGMathBench, much lower on other text-only mathematical benchmarks. 2. All LLMs evaluated exhibit high reasoning gap with Robustness Efficiency (RE, the ratio between $\Delta$ and EAcc) ranging from 20.78% to 196.6%, pinpointing the inconsistencies of current LLMs in solving problems with variable disturbance. 3. There remains a significant discrepancy among closed-source LLMs and open-source LLMs (even specialized mathematical LLMs). Among open-source LLMs, only Qwen-2-Math-72B-Instruct and Mistral-Large-Instruct have comparable performance with GPT-4o. 4. The average EAcc varies by subject, with Arithmetic scoring 62.8%. In contrast, Abstract Algebra, Differential Equations, and Financial Mathematics have average EAccs of less than 10%. 5. An error analysis of OpenAI-o1-mini's performance reveals that calculation errors are a major concern. Even the same problem presented in different randomized versions can lead to varying types of errors.

Table 1: Comparison of various benchmarks. "#Types" indicates the number of answer types in the dataset. "#Subjects" specifies the number of mathematical subjects covered. "Dynamic" denotes whether the dataset is dynamic or static. "#Test" shows the number of test examples in the dataset, while "#College" refers to the number of test examples at the college level.

| Dataset | Level | #Types | #Subjects | Dynamic | #Test | #College |
|---------|-------|--------|-----------|---------|-------|----------|
| GSM8K | Elementary | 1 | - | ✗ | 1,319 | 0 |
| MATH | Competition | 3 | 7 | ✗ | 5,000 | 0 |
| MMLU-Math | All | 1 | - | ✗ | 844 | 116 |
| TAL-SCQ | K12 Math | 1 | - | ✗ | 1,496 | 0 |
| AGIEval-SAT-Math | High School | 2 | - | ✗ | 102 | 0 |
| AGIEval-Math | Competition | 2 | - | ✗ | 938 | 0 |
| CollegeMath | College | 3 | 7 | ✗ | 2,818 | 2,818 |
| MathBench | All | 1 | 5 | ✗ | 1781 | 466 |
| GSM1K | Elementary | 1 | - | ✓ | 1,250 | 0 |
| FN-EVAL | Competition | 3 | 7 | ✓ | 2,060 | 0 |
| VarBench-Math | Elementary | 1 | - | ✓ | 1,319 | 0 |
| LiveBench-Math | Competition | 2 | - | ✓ | 232 | 0 |
| UGMathBench | College | **10** | **16** | ✓ | **5,062** | **5,062** |

## 2 RELATED WORK

**Mathematical Benchmarks**. Mathematical reasoning is increasingly vital for assessing the fundamental reasoning capabilities of LLMs (Ahn et al., 2024). Several math-related datasets have been proposed in this area (Koncel-Kedziorski et al., 2016; Amini et al., 2019; Hendrycks et al., 2021a; Cobbe et al., 2021; Hendrycks et al., 2021b; Chen et al., 2022). Among these, GSM8K (Cobbe et al., 2021) and MATH (Hendrycks et al., 2021b) are the most representative datasets for elementary and high school-level math reasoning, respectively. However, as modern LLMs become increasingly powerful, these benchmarks lack sufficient challenge forlatest LLMs. Notably, o1 (OpenAI, 2024b) achieves 94.8% accuracy on MATH, which was previously considered highly complex. To better assess the mathematical reasoning abilities of current LLMs, some researchers create variants of existing benchmarks (Shi et al., 2023; Chen et al., 2024; Li et al., 2024b; Xu et al., 2024b), while others propose new, more challenging math reasoning benchmarks (Chen et al., 2023; Wang et al.,

2023a; Collins et al., 2024; Tang et al., 2024; Liu et al., 2024; Wang et al., 2024). CollegeMath (Tang et al., 2024) covers several college-level mathematics subjects with limited answer types. In contrast, our UGMathBench encompasses a broader range of subjects, answer types, and test examples. In addition, there are also several cross-modality math-related datasets (Chen et al., 2021; Lu et al., 2023; Xu et al., 2024a; Yue et al., 2024a; He et al., 2024b;a; Huang et al., 2024; Yue et al., 2024b).

**Dynamic Benchmarks for Mathematical Reasoning**. Test set contamination, wherein benchmark test data appear in a newer model's training set, significantly challenges fair LLM evaluation by artificially inflating performance (Deng et al., 2024; Dong et al., 2024; Golchin & Surdeanu, 2023; Roberts et al., 2023). Since pretraining data often involve large corpora scraped from the Internet, any static benchmark risks data contamination (Zhang et al., 2024; Qian et al., 2024). To mitigate this, recent benchmarks maintain private test sets (Zhang et al., 2024; Huang et al., 2024), requiring anyone who wishes to evaluate their models to submit predictions for centralized processing before publishing results on their leaderboards. However, this process can be inefficient and lacks transparency for error analysis (Qian et al., 2024). An alternative is releasing dynamic benchmarks that are periodically updated (Srivastava et al., 2024; Qian et al., 2024; White et al., 2024). For example, Srivastava et al. (2024) have functionalized a subset of the MATH dataset to regenerate new versions of the test set by reassigning variable values. In this vein, our UGMathBench is a dynamic benchmark featuring different sampled values for variables by setting distinct random seeds. Currently, we release three snapshots for each question in UGMathBench and plan to release new versions if leading open-source LLMs reach accuracy saturation.

## 3 THE UGMATHBENCH BENCHMARK

### 3.1 UGMATHBENCH OVERVIEW

We introduce the UGMathBench, a dynamic undergraduate-level mathematical reasoning benchmark designed to thoroughly and robustly assess the mathematical reasoning ability of LLMs. UGMathBench enables fair evaluation through randomized versions of single problems. Unlike GSM1K (Zhang et al., 2024), our test set labels are publicly available, facilitating efficient evaluation and effective error analysis. UGMathBench covers fifteen core subject areas in undergraduate-level mathematics, including single-variable calculus, multivariable calculus, differential equations, probability, and more, encompassing a total of 111 specific topics (details in Appendix A.2). UGMathBench comprises a set of 5,062 problems

Table 2: Benchmark Statistics

| Statistic | Number |
| --- | --- |
| Total Problems | 5562 |
| Number of Versions | x 3 |
| Total Subjects/topics | 16/111 |
| Total Answer Types | 10 |
| Total Difficulty Level | 6 |
| Average Problem Tokens | 122.63 |
| Average Number of Answers | 2.77 |

in 3 different randomized snapshots with 10 different answer types (see Appendix A.3). These answer types range from atomic types (e.g., numerical value, expression) to compound types (e.g., multiple answers in ordered or unordered lists), setting UGMathBench apart from many other math-related benchmarks that focus primarily on a single answer with an atomic type. We randomly select 100 problems to examine student performance using our grading system' records, with each problem being completed by varying numbers of students ranging from 99 to 1,537. The average accuracy on the first attempt is 56.5%, while the average accuracy on the final attempt increased to 96.1%.

### 3.2 UGMATHBENCH CREATION

Our UGMathBench creation process has three distinct phases: data collection, data cleaning & deduplication, and answer type annotation.

**Data Collection**. The dataset for UGMathBench is carefully compiled from the online grading system of our institute's undergraduate courses (see Appendix B.1). All problems in our system are generated by programs that specify particular variable values to ensure correctness and maintain the same solution (see Appendix A.1). We gather all mathematics-related problems, resulting in 16 subjects and 111 topics in total. To prevent student cheating, our grading system offers randomized versions of most problems (see Figure 1), similar to the variable disturbance approach in Qian et al. (2024). To create a dynamic benchmark, we exclude static problems without randomized versions, as well as those containing images, ensuring a text-only reasoning benchmark. The collected problems are originally in HTML format.

**Data Cleaning and Deduplication**. After collecting problems in the HTML format, we utilize the `bs4`[1] and `re`[2] Python packages to convert them into Latex. Since no conversion process is flawless, we manually verify the converted LaTeX files against the original HTML files. The latex files are then further organized into the format shown in Figure 1. After converting and cleaning all the problems, we perform deduplication within each subject based on embeddings generated by `text-embedding-ada-002` to remove duplicated problems (see Appendix B.2). The thresholds and the number of questions that are filtered out are given in Table 10.

**Answer Type Annotation**. Meta-information (e.g. subject, topic, subtopic, difficulty level as shown in Figure 1) is stored in our grading system and is easily extracted along with the LaTeX files. The primary task is determining the answer types. Problems requiring definitive responses are largely categorized into two main classes: atomic and compound. Questions with a single required answer fall into the atomic type, while questions with multiple answers are classified as compound, represented by a list of atomic answers separated by commas. The atomic type can be further classified into eight types, and the compound answer lists can be either ordered or unordered, with each atomic answer fitting one of the aforementioned eight types. Simple examples of each type are provided in Table 3, and detailed definitions are available in Appendix A.3.

Table 3: Examples of eight atomic answer types.

| Type | Example |
|---|---|
| Numerical Value | $\pi/4$ |
| Expression | $x^2 + 1$ |
| Equation | $x^2 + y^2 = 1$ |
| Interval | $(-\infty, -1]$ |
| True/False | Yes |
| MC with single answer | A |
| MC with multiple answers | ACF |
| Open-Ended | h(1-x) |

## 3.3 EVALUATION METRICS

We denote the set of test examples in UGMathBench by $\mathcal{D}$ with a specific test example denoted as $e_i$, where $i$ represents the index of the example. Each example $e_i$ consists of questions presented in different randomized versions: $q_i^1, q_i^2, ..., q_i^V$, where $V$ is the total number of versions[3]. The corresponding ground-truth answers for these versions $q_i^1, q_i^2, ..., q_i^V$ are denoted by $a_i^1, a_i^2, ..., a_i^V$. The answer generated by an LLM $\mathcal{M}$ for a specific version of the question in the $i$-th test example is denoted by $\mathcal{M}(q_i^v)$. Inspired by Srivastava et al. (2024), we define the following metrics to evaluate the true mathematical reasoning ability of LLM $\mathcal{M}$ in UGMathBench.

**Accuracy of Version v Acc$_\mathbf{v}$** is defined as the average accuracy of model $\mathcal{M}$ on the set of questions with version $v$ in $\mathcal{D}$:

$$\text{Acc}_v = \frac{\sum_{i=1}^{|\mathcal{D}|} \mathbb{I}[\mathcal{M}(q_i^v) = a_i^v]}{|\mathcal{D}|},$$

where $\mathbb{I}$ is an indicator function and $|\mathcal{D}|$ denotes the number of examples in UGMathBench. It assesses the performance of an LLM on the specific version $v$ from UGMathBench.

**Average Accuracy AAcc** is defined as the mean of all Acc$_v$:

$$\text{AAcc} = \frac{\sum_{v=1}^{V} \text{Acc}_v}{V}.$$

This metric evaluates the performance across all versions of the questions.

**Effective Accuracy EAcc** is defined as the accuracy in solving a test example $e_i$ across all its $V$ versions:

$$\text{EAcc} = \frac{\sum_{i=1}^{|\mathcal{D}|} \mathbb{I}[\mathcal{M}(q_i^v) = a_i^v, \forall v \in \{1, 2, ..., V\}]}{|\mathcal{D}|}.$$

If a model is able to solve a test case using proper reasoning, it should correctly solve this problem for all randomized versions. Thus, effective accuracy measures the fraction of test cases correctly solved across all versions $V$. It measures true reasoning of test cases in UGMathBench.

**Reasoning Gap $\Delta$** is defined as the percentage decrease between AAcc and EAcc. It provides a measure of the robustness of reasoning, with $\Delta = 0$ being true reasoning with high robustness.

---

[1] https://pypi.org/project/beautifulsoup4/

[2] https://docs.python.org/3/library/re.html

[3] Currently, V=3 and we plan to release more versions in the future.

**Robustness Efficiency (RE)** is defined as the ratio of the Reasoning Gap ($\Delta$) to the EAcc, expressed as RE $= \Delta$/EAcc. This metric evaluates the extent of the reasoning gap relative to the model's effective reasoning ability (i.e., EAcc). RE captures robustness by taking the effectiveness of mathematical reasoning into account , with lower values indicating superior performance in adapting to variations across different versions of problems in UGMathBench. Achieving a higher EAcc and a lower $\Delta$ results in a more favorable (lower) RE, reflecting improved robustness relative to "true" reasoning ablility of LLMs.

# 4 EXPERIMENTS

## 4.1 EXPERIMENTAL SETUP

**Evaluated LLMs**. Our evaluation covers 23 leading LLMs, including closed-source commercial LLMs and open-source LLMs. Based on our UGMathBench, we provide a thorough evaluation of the mathematical reasoning capabilities of current LLMs. The evaluated LLMs are listed below:

- For proprietary LLMs, we select OpenAI-o1-mini (OpenAI, 2024b), GPT4o (OpenAI, 2024a), GPT4o-mini (OpenAI, 2024a), and Claude-3-Opus (Anthropic, 2024).
- For open-source general-purpose LLMs, we evaluated the LLaMA-3-Instruct series (8B, 70B) (AI@Meta, 2024), Qwen2-Instruct (7B, 72B)(Yang et al., 2024a), Yi-1.5-Chat (6B, 9B, 34B) (AI et al., 2024), Mistral-7B-Instruct (Jiang et al., 2023), Mistral-Nemo-Instruct-2407 (Mistral, 2024c), Mistral-Small-Instruct-2409 (Mistral, 2024d), Mistral-Large-Instruct-2407 (Mistral, 2024b), DeepSeek-MOE-16B-Chat (Dai et al., 2024), and DeepSeek-V2-Lite-Chat (DeepSeek-AI, 2024).
- We also include some specialized math LLMs: DeepSeekMath-7B (-RL, -Instruct) (Shao et al., 2024), Qwen2-Math (7B, 72B)(Yang et al., 2024b), Mathstral-7B (Mistral, 2023), and NuminaMath-7B-CoT (Beeching et al., 2024).

Details of these LLMs are provided in Appendix C.1.

**Evaluation Settings**. We employ $\text{Acc}_v$ to evaluate the average performance of version $v$, AAcc to measure the average performance across all versions, EAcc to quantify true reasoning, and reasoning gap $\Delta$ to assess the robustness of reasoning (see Section 3.3). To remove the effect of sensitivity of few-shot prompts (Lu et al., 2022; Ma et al., 2023), all our experiments use zero-shot prompts, tailored to different answer types for better answer extraction and rule-based matching. Detailed prompts are given in Appendix C.2. We use vLLM[4] to speed up the evaluation process. To maintain consistency in evaluations and facilitate reproduction, we set the maximum output length to 2,048 tokens and employ a greedy decoding strategy with temperature 0.

## 4.2 MAIN RESULTS

The overall experiment results are shown in table 4. We have the following key observations:

**UGMathBench is a challenging benchmark for evaluating the mathematical reasoning capabilities of LLMs.** Even LLMs with the most advanced reasoning abilities, OpenAI-o1 (mini version), achieve only 56.3% EAcc on UGMathBench, while most open-source LLMs, including most specialized mathematical models, struggle to reach a 30% EAcc. Compared to commonly used mathematics benchmarks like MATH (Hendrycks et al., 2021b), UGMathBench proves to be more challenging. For instance, OpenAI-o1-mini achieves 90% on MATH (v.s. 56.3% on UGMathBench).

**Even leading LLMs still have inconsistencies when solving problems with multiple versions**. LLMs with an AAcc greater than 20% display a reasoning gap $\Delta$ exceeding (or near) 10%. All LLMs demonstrate extremely high RE on UGMathBench, with values ranging from 20.78% to 196.6%. Among the five models with the lowest RE, three of them are from OpenAI (OpenAI-o1-mini: 20.78%; GPT-4o: 20.89%; Mistral-Large-Instruct: 24.36%; Qwen2-Math-72B-Instruct: 24.39%; GPT-4o-mini: 27.87%). These results pinpoint the limitaions of current LLMs and urge us to develop ture "large reasoning models" with high EAcc and $\Delta = 0$.

---

[4]https://github.com/vllm-project/vllm

Table 4: **Main Results on UGMathBench** (all figures are in %). Models are classified into three categories according to their purpose and origin. The best results within each column are **bolded** and the best results of open-source Chat LLMs within a similar parameter size group are underlined.

| Models | Acc$_1$ | Acc$_2$ | Acc$_3$ | AAcc | EAcc | $\Delta$ | RE |
|---|---|---|---|---|---|---|---|
| *Closed-source LLMs* | | | | | | | |
| OpenAI-o1-mini-2024-09-12 | **68.02** | **67.95** | **68.04** | **68.00** | **56.30** | 11.70 | **20.78** |
| GPT-4o-2024-08-06 | 59.92 | 60.79 | 60.41 | 60.37 | 49.94 | 10.43 | 20.89 |
| GPT-4o-mini-2024-07-18 | 51.58 | 53.14 | 52.61 | 52.44 | 41.01 | 11.43 | 27.87 |
| Claude-3-Opus-20240229 | 48.62 | 50.32 | 49.47 | 49.47 | 37.00 | 12.47 | 33.69 |
| *Open-source Chat LLMs* | | | | | | | |
| Yi-1.5-6B-Chat | 25.33 | 26.83 | 26.59 | 26.25 | 15.23 | 11.02 | 72.33 |
| Mistral-7B-Instruct | 10.19 | 11.16 | 10.33 | 10.56 | 4.44 | 6.12 | 137.6 |
| Qwen2-7B-Instruct | 35.60 | 37.30 | 36.23 | 36.38 | 25.15 | 11.23 | 44.65 |
| LLaMA3-8B-Instruct | 16.00 | 17.01 | 16.63 | 16.55 | 8.91 | 7.64 | 85.74 |
| Yi-1.5-9B-Chat | 33.72 | 34.29 | 34.85 | 34.29 | 21.12 | 13.17 | 62.36 |
| Mistral-Nemo-Instruct-2407 | 24.53 | 25.62 | 25.09 | 25.08 | 15.43 | 9.65 | 62.57 |
| DeepSeek-MOE-16B-Chat | 5.59 | 5.85 | 5.97 | 5.80 | 1.96 | **3.85** | 196.6 |
| DeepSeek-V2-Lite-Chat | 12.82 | 13.67 | 12.76 | 13.08 | 5.69 | 7.39 | 130.0 |
| Mistral-Small-Instruct-2409 | 40.10 | 40.52 | 40.04 | 40.22 | 28.84 | 11.38 | 39.45 |
| Yi-1.5-34B-Chat | 37.08 | 38.11 | 37.65 | 37.61 | 24.34 | 13.28 | 54.55 |
| LLaMA3-70B-Instruct | 33.25 | 34.35 | 33.27 | 33.62 | 23.27 | 10.35 | 44.48 |
| Qwen2-72B-Instruct | 47.49 | 48.56 | 47.23 | 47.76 | 35.78 | 11.98 | 33.50 |
| Mistral-Large-Instruct-2407 | 55.91 | 56.16 | 55.97 | 56.01 | 45.04 | 10.97 | 24.36 |
| *Specialized Mathematical LLMs* | | | | | | | |
| DeepSeek-Math-7B-Instruct | 23.61 | 24.87 | 23.19 | 23.89 | 13.61 | 10.28 | 75.52 |
| DeepSeek-Math-7B-RL | 28.66 | 29.97 | 28.94 | 29.19 | 19.24 | 9.95 | 51.71 |
| NuminaMath-7B-CoT | 29.32 | 30.07 | 30.01 | 29.80 | 18.81 | 10.99 | 58.44 |
| Mathstral-7B-v0.1 | 28.57 | 28.47 | 28.51 | 28.51 | 17.94 | 10.58 | 58.96 |
| Qwen2-Math-7B-Instruct | 43.01 | 44.13 | 44.05 | 43.73 | 32.46 | 11.27 | 34.73 |
| Qwen2-Math-72B-Instruct | 56.95 | 57.05 | 57.09 | 57.03 | 45.85 | 11.18 | 24.39 |

**There remains a significant discrepancy between closed-source models and open-source LLMs.** OpenAI-o1-mini achieves the best results across Acc$_i$, $i = 1, 2, 3$, as well as in AAcc and EAcc. However, the most powerful open-source LLM evaluated, Qwen2-Math-72B-Instruct, still falls short: it shows a 10.97% lower AAcc and a 10.45% lower EAcc compared to OpenAI-o1-mini. The best-performing open-source chat model is Mistral-Large-Instruct-2407, which ranks second among all open-source LLMs. Only 2 out of 19 open-source LLMs exceed GPT-4o-mini in terms of EAcc, and only 3 out of 19 have an EAcc comparable to Claude-3-Opus. Additionally, more than half of the open-source LLMs (10 out of 19) have an EAcc smaller than 20%.

## 5 ANALYSIS

In this section, we conduct an in-depth analysis of the performance of the 23 LLMs evaluated on UGMathBench by investigating the following research questions: 1. What is the relationship between EAcc and Acc$_v$? (Section 5.1) 2. How do model size and model series influence performance on UGMathBench? (Section 5.2) 3. How do LLMs perform across different subjects, difficulty levels, and, different topics on UGMathBench? (Section 5.3) 4. What are the typical response errors made by the best-performing LLM (OpenAI-o1-mini), and how are they distributed? (Section 5.4)

### 5.1 RELATIONSHIP BETWEEN EACC AND ACC$_v$

To investigate the relationship between EAcc and Acc$_v$, scatter plots of EAcc against each Acc$_v$ are shown in Figure 2. From Figure 2, we have the following conclusions:

**All LLMs fall below the diagonal lines**. Each LLM evaluated is represented as a point in the subfigures of Figure 2, plotted on the axes of (Acc$_v$, EAcc). Although LLMs exhibit small variations in accuracy across different versions, they consistently demonstrate a lower EAcc than Acc$_v$, which suggests that the accuracy of individual versions is insufficient for assessing the reasoning capabilities of LLMs. By considering EAcc alongside accuracy, we can gain a better understanding of how LLMs

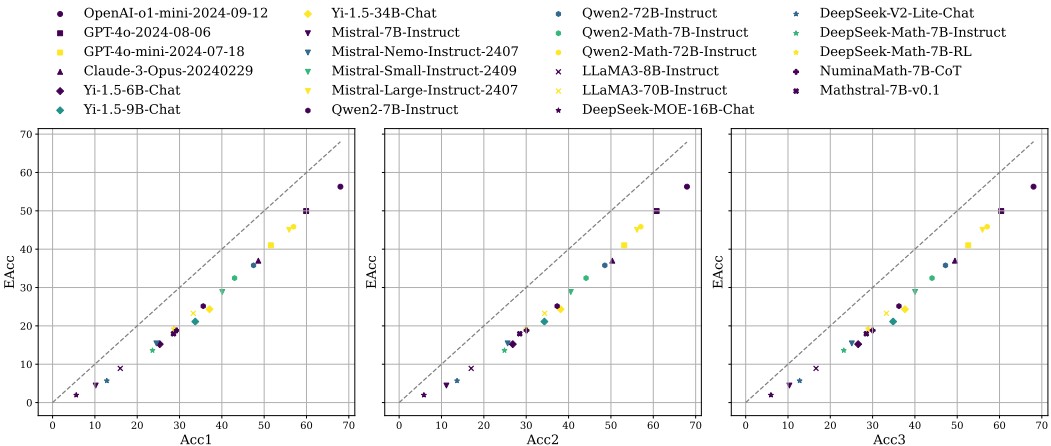

Figure 2: EAcc v.s. Acc$_v$ on UGMathBench.

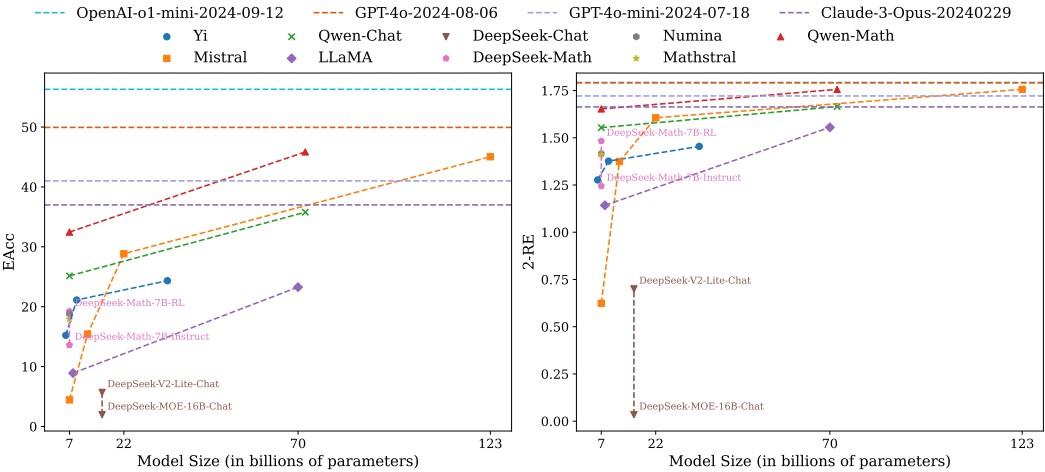

Figure 3: Left: EAcc v.s. Model Size. Right: 2-RE v.s. Model Size. The comparison chart of performance versus performance (EAcc and RE) on UGMathBench for all LLMs evaluated, with models from the same series connected by lines of the same color. The horizontal dotted lines represent the score of close-source LLMs.

perform when solving problems that have different randomized versions. The discrepancy between EAcc and Acc$_v$ highlights a new inconsistency mode (Ahn et al., 2024) of current LLMs: they may become inconsistent in their answers when the problem is slightly altered.

**There is an apparent trend that a high EAcc consistently leads to a high Acc$_v$.** A model with high EAcc is more effective in handling variable disturbances, resulting in high accuracy for each version. However, as EAcc becomes increasingly large, the difference between Acc$_v$ and EAcc tends to increase until it stabilizes around 10%.

## 5.2 THE EFFECT OF MODEL SIZE AND MODEL SERIES

Figure 3 has shown how EAcc and 2-RE changes with the parameter size. We can observe that:

**LLMs within the same series have shown steady improvement as the parameter size increases.** When the model size increases from 7B to around 100B, EAcc substantially improve and RE steadily decrease for Qwen-Chat, Qwen-Math, Mistral, Deepseek-Chat, LLaMA-3-Instruct, and Yi-Chat series, indicating a steady improvement in performance in effectiveness and robustness of mathematical reasoning.

**Specialized mathematical LLMs typically outperform their general-purpose counterparts**. For example, the Qwen2-Math series achieves significantly higher EAcc and lower RE than its general-purpose chat LLMs with the same model size. Among all 7B specialized mathematical LLMs, Qwen2-Math-7B-Instruct ranks first surpassing DeepSeek-Math-RL (second best) by a large margin.

## 5.3 Performance Across Different Subjects and Difficulty Levels

Figure 4a shows the average EAcc of different models in each subject. Figure 4b shows the averaged EAcc of all subjects with respect to different levels of difficulty. The detailed performances across different subjects and topics for each model can be found in Appendix D and F.

**The average EAcc varies across different subjects**. LLMs are effective at solving Arithmetic problems, achieving 62.8% EAcc. In addition to Arithmetic, LLMs are also adept at Algebra, Combinatorics, and Complex Analysis (over 30% average EAcc). The three least effective areas are Abstract Algebra, Differential Equations, and Financial Mathematics, which typically require challenging domain knowledge (less than 10% average EAcc).

**In general, the averaged EAcc decreases as the level increases.** OpenAI's o1-mini is the strongest among all models and wins by a larger margin as the level increases. Mistral-Large/Small-Instruct and Qwen2 series are the most competitive open-source models on our benchmark, with EAcc comparable to leading commercial models such as GPT-4o. However, as the level of difficulty increases, they still lag behind GPT-4o, suggesting the gap between the leading open-source LLMs and proprietary LLMs in solving difficult math problems.

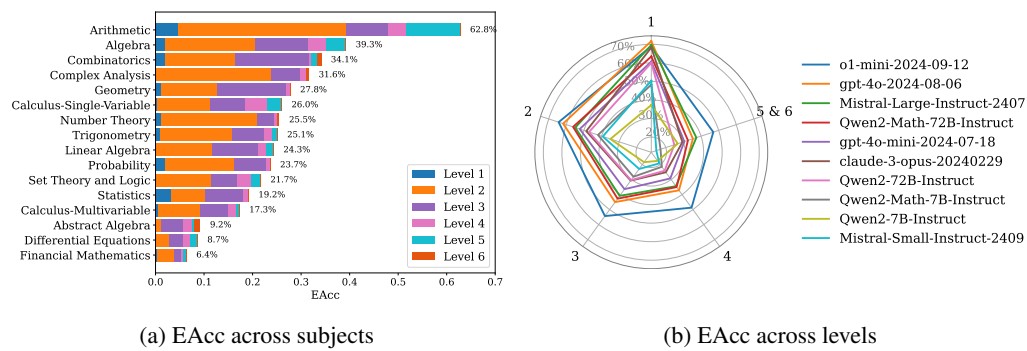

(a) EAcc across subjects                           (b) EAcc across levels

Figure 4: Relationship between EAcc, subject, and level of difficulty. (a) EAcc of different subjects, averaged across all models. Each bar consists of several segments with colors indicating their corresponding difficulty level. Notice that the length of each color segment only indicates its proportion within all problems of all levels within that subject, and is not comparable between levels or subjects. (b) EAcc of different levels, averaged across all subjects. Only models with top-10 EAcc are included for brevity. Levels 5 and 6 are combined since level 6 has few samples.

## 5.4 Error Analysis

We perform a comprehensive error analysis on OpenAI-o1-mini by randomly selecting 100 problems, each having at least one incorrectly solved version (yielding a total of 300 versions). As shown in Figure 5a, there are 231 incorrect versions, and OpenAI-o1-mini failed to solve 56% of the problems across all versions. We then categorize these errors into six types, as illustrated in Figure 5b. Calculation errors, including both numerical and expression errors, represent the largest category, with several examples provided in Appendix E. We find that OpenAI-o1-mini tends to streamline its outputs to avoid generating too long responses, sometimes leading to erroneous results. Additionally, we encounter some "bad questions" that primarily arise due to overly complex structures (e.g. containing long tables) or inadequately described (e.g. undefined variables in previous problems). This is because our homework grading system (see Appendix B.1) is designed for students with a user-friendly interface, and some problems may not be suitable for LLM to solve. In our sample of 300 versions, 19 were identified as "bad problems," giving us an estimated occurrence of approximately 2.7% in our UGMathBench. These problems do not impact our main claims, as no LLMs are able

to solve these "bad questions." We will refine these types of problems to make them more suitable for LLM evaluation in the future. No answer cleaning process is perfect, and improved evaluation codes continue to be released in MATH (Hendrycks et al., 2021b). We will also actively update our evaluation repository to improve its quality. Inspired by recent model-based evaluation efforts (Gao et al., 2024; Xu et al., 2025a), we have updated our code repository to integrate the model-based evaluation introduced by Xu et al. (2025a). Updated results will be made available there.

Notably, we have found that even when OpenAI-o1-mini solves a problem incorrectly among all its randomized versions, the error types can be different. As shown in Figure 5a, there are around 16.1% such inconsistent errors among the 100 problems sampled.

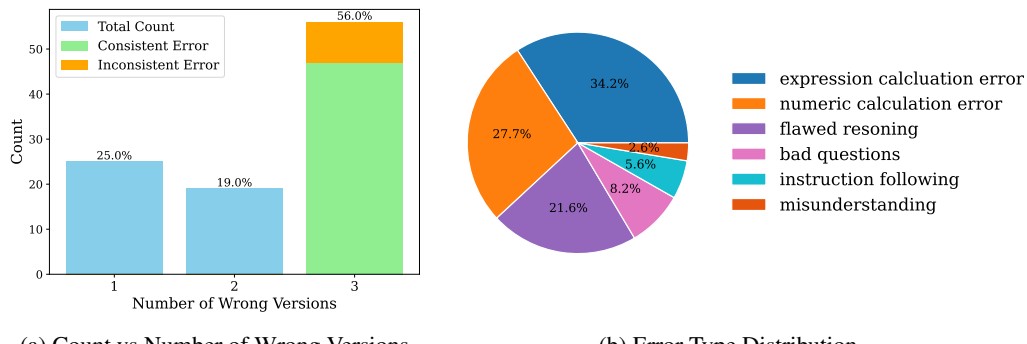

(a) Count vs Number of Wrong Versions         (b) Error Type Distribution

Figure 5: Error Analysis of OpenAI-o1-mini on UGMathBench.

## 5.5 FURTHER ANALYSIS

**About Self-Improvement**. To examine how LLMs perform with refinement on UGMathBench, we conducted experiments using Progressive-Hint Prompting (PHP) (Zheng et al., 2023) (see Appendix G). Detailed results can be found in Table 32 in Appendix G. Although PHP improves EAcc and AAcc for most LLMs, the enhancements are not significant, indicating considerable room for future development. The fine-grained results in Table 33 suggest that the impact of refinement for GPT-4o varies across different subjects. For instance, PHP improves GPT-4o's performance in abstract algebra by 7.14%, yet reduces its performance in probability by 2.08% in terms of EAcc. Our UGMathBench serves as an excellent testing ground for future research into refinement methods for solving undergraduate-level mathematics with LLMs.

**Reasoning Gap and Test Set Contamination**. To explore how models specifically overfitting to a particular variation affect the reasoning gap, we mixed a portion of the test set from one version with MetaMathQA (Yu et al., 2023) and then conducted supervised fine-tuning (SFT) Llama-3-8B on this data. Details of the SFT process are provided in Appendix H, and the results are presented in Table 5. As the proportion of the test set included in the training

Table 5: Effects of Overfitting.

| Proportion | 5% | 10% | 15% | 20% |
|---|---|---|---|---|
| Δ | 3.43 | 3.80 | 4.50 | 4.64 |

data increases, the reasoning gap (Δ) also becomes more pronounced. This study serves as an initial investigation into test set contamination during the SFT stage. It's worth noting that contamination at the pre-training stage is also a significant area of interest (Razeghi et al., 2022; Jiang et al., 2024).

## 6 CONCLUSION

Current mathematical benchmarks are often inadequate, lacking comprehensive coverage of undergraduate-level math problems or being susceptible to test-set contamination. To fill these gaps, we propose UGMathBench, a diverse and dynamic benchmark for undergraduate-level mathematical reasoning. Our fine-grained analysis has pointed out the potential inconsistencies when LLMs encounter problems with slightly different versions. We hope that our UGMathBench can contribute to future development of "true" reasoning LLMs.

ACKNOWLEDGMENTS

This work was partially supported by a grant from the Research Grants Council of the Hong Kong Special Administrative Region, China (Project Reference Number: AoE/E-601/24-N). We would like to thank all reviewers for their helpful suggestions in improving this paper.

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

## LIMITATIONS

This work has several limitations. First, UGMathBench focuses on text-only reasoning, whereas some undergraduate-level math problems require images for their solutions. Developing a multimodal benchmark for undergraduate-level mathematics will be future work. Second, UGMathBench is designed as an English-language benchmark. Extending UGMathBench to support multiple languages could be an interesting avenue for future research. Third, the number of problems in certain subjects is limited. Expanding these subjects would be valuable.

## A    DETAILED STATISTICS OF UGMATHBENCH

### A.1    INTRINSIC MECHANISM OF DYNAMIC PROBLEMS

A key feature of our UGMathBench is its dynamic nature. In this appendix, we detail how this is achieved. All problems in our homework grading system (see Appendix B.1) are stored as programs written in Program Generation, an established programming language for mathematics. These programs strictly specify conditions to ensure that the generated variations of each problem do not fundamentally alter their nature, required solution approach, difficulty level, or underlying knowledge points. One such program is shown in Listing 1, and two different versions of this problem it generates are given in Figure 7. The relationship between the first and second variables is defined by $\$expnt = -1 + 2 * a;$, which maintains the consistency of the concepts, techniques, and solutions involved in different versions of each problem.

### A.2    DISTRIBUTION OF PROBLEMS

Our UGMathBench covers various subjects in undergraduate-level mathematics. The detailed topics and the number of subtopics of each topic across different subjects are listed in Table 6 and 7. There are 111 topics and 583 subtopics across 16 subjects in total. Furthermore, the distribution information of our benchmark on different subjects and difficulty level is presented in Table 8. Note that there are problems with missing difficulty level in our online homework grading system (see Appendix B.1) and we remain as is for consistency. The keywords of our UGMathBench are shown in Figure 6. The detailed results across different subjects and topics are discussed in Appendix D and F.

### A.3    ANSWER TYPES

By carefully reviewing a large collection of problems and referring to various past benchmarks (He et al., 2024a; Huang et al., 2024), we classify all answers to be two main categories: atomic and compound. There are 8 atomic types and 2 compound types. Each compound type is composed of a list of atomic ones. These types are designed to encompass a wide range of problems. Detailed definitions for each answer type can be found in Table 9.

## B    UGMATHBENCH CREATION

### B.1    DATA SOURCE

UGMathBench originates from questions in the online homework grading system of our institute, utilizing WebWork[5], an open-source online platform licensed under GNU. Widely employed for assigning mathematics and science homework in educational settings, WebWork benefits from collaborative contributions by educators across various institutions.

Each question in WebWork is tagged with keywords related to concepts and difficulty level based on Bloom's taxonomy[6], which helps simplify statistical analysis and cognitive assessment. To prevent cheating from each other, WebWork is able to generate tailored problem sets with different random seeds, making it popular among educational institutions.

---

[5]https://www.webwork.org
[6]https://webwork.maa.org/wiki/Problem_Levels

Table 6: Topics of each subject and corresponding number of subtopics included in UGMathBench.

| Subject | Topics | # Sub-Topics |
|---|---|---|
| Arithmetic | Integers | 11 |
| | Fractions/rational numbers | 12 |
| | Decimals | 9 |
| | Percents | 3 |
| | Irrational numbers | 1 |
| | Other bases | 3 |
| | Units | 2 |
| Algebra | Algebra of real numbers and simplifying expressions | 8 |
| | Absolute value expressions and functions | 3 |
| | Properties of exponents, rational exponents and radicals | 2 |
| | Cartesian coordinate system | 4 |
| | Factoring | 5 |
| | Functions | 8 |
| | Transformations of functions and graphs | 6 |
| | Linear equations and functions | 9 |
| | Quadratic equations and functions | 8 |
| | Operations on polynomial and rational expressions | 7 |
| | Polynomial equations and functions | 7 |
| | Variation and power functions | 5 |
| | Systems of equations and inequalities | 1 |
| | Functions with fractional exponents and radical functions | 1 |
| | Rational equations and functions | 6 |
| | Inverse functions | 3 |
| | Exponential and logarithmic expressions and functions | 8 |
| | Finite sequences and series | 4 |
| | Conic sections | 3 |
| Set theory and logic | Operations on sets | 5 |
| | Relations between sets | 2 |
| | Functions | 2 |
| | Propositional logic | 4 |
| | First order logic | 3 |
| | Pattern matching | 1 |
| Trigonometry | Geometric and algebraic foundations for trigonometry | 3 |
| | Trigonometric functions | 8 |
| | Triangle trigonometry | 4 |
| | Analytic trigonometry | 7 |
| | Polar coordinates & vectors | 2 |
| Combinatorics | Counting | 8 |
| | Recurrence relations | 3 |
| Geometry | Shapes | 4 |
| | Circle geometry | 1 |
| | Vector geometry | 7 |
| Calculus single-variable | Calculus of vector valued functions | 6 |
| | Concepts for multivariable functions | 6 |
| | Differentiation of multivariable functions | 7 |
| | Integration of multivariable functions | 9 |
| | Vector fields | 1 |
| | Vector calculus | 6 |
| | Fundamental theorems | 4 |
| Calculus multivariable | Limits and continuity | 14 |
| | Differentiation | 13 |
| | Applications of differentiation | 18 |
| | Integrals | 4 |
| | Techniques of integration | 8 |
| | Applications of integration | 16 |
| | Infinite sequences and series | 18 |
| | Parametric | 6 |
| | Polar | 3 |
| Linear Algebra | Systems of linear equations | 7 |
| | Matrices | 8 |
| | Matrix factorizations | 5 |
| | Euclidean spaces | 8 |
| | Abstract vector spaces | 7 |
| | Eigenvalues and eigenvectors | 5 |
| | Inner products | 6 |
| | Linear transformations | 6 |
| | Determinants | 3 |
| Number Theory | Divisibility | 4 |
| | Congruences | 5 |
| | Diophantine equations | 1 |
| Financial Mathematics | Annuities | 5 |
| | Bonds | 3 |
| | Equations of value | 2 |
| | Interest | 6 |
| | Options | 5 |
| | Expected and contingent payments | 2 |
| | Equities | 2 |

Table 7: Topics of each subject and corresponding number of subtopics included in UGMathBench (Con't).

| Subject | Topics | # Sub-Topics |
|---|---|---|
| Probability | Sample Space | 6 |
| | Random variables | 5 |
| | Discrete distributions | 6 |
| | Continuous distributions | 5 |
| | Laws, theory | 2 |
| | Several variables | 2 |
| Statistics | Experimental design | 1 |
| | Sample survey methods | 2 |
| | Exploratory data analysis/descriptive statistics | 5 |
| | Sampling distributions | 3 |
| | Confidence intervals | 9 |
| | Hypothesis tests | 13 |
| | Simple linear regression | 7 |
| | Time series | 2 |
| | Point estimation | 1 |
| | Multiple regression | 5 |
| | Generalized linear methods | 1 |
| Complex Analysis | Arithmetic | 8 |
| | Complex equations | 1 |
| | Complex plane | 1 |
| | Complex functions | 2 |
| | Analytic functions | 3 |
| | Series and residues | 4 |
| Differential Equations | Introductory concepts | 2 |
| | First order differential equations | 13 |
| | Higher order differential equations | 9 |
| | Laplace transforms | 6 |
| | Systems of differential equations | 4 |
| | Numerical methods | 1 |
| | Series solutions | 3 |
| | Partial differential equations | 7 |
| Abstract Algebra | Groups | 7 |
| | Rings | 4 |
| | Fields and polynomials | 1 |
| Total count | 111 | 583 |

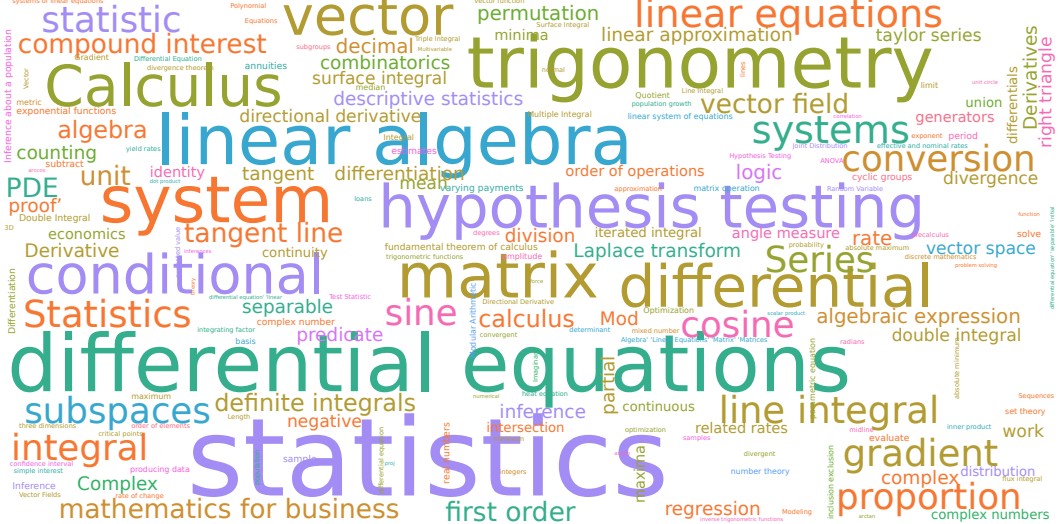

Figure 6: Word Cloud of Keywords in UGMathBench.

Table 8: Statistics of UGMathBench across different subjects and difficulty levels.

| | Level 1 | Level 2 | Level 3 | Level 4 | Level 5 | Level 6 | All |
|---|---|---|---|---|---|---|---|
| Arithmetic | 22 | 179 | 47 | 21 | 53 | 0 | 322 |
| Algebra | 25 | 233 | 176 | 80 | 73 | 0 | 583 |
| Set theory and logic | 0 | 35 | 11 | 12 | 9 | 0 | 69 |
| Trigonometry | 5 | 86 | 44 | 17 | 25 | 0 | 178 |
| Combinatorics | 3 | 28 | 47 | 4 | 5 | 1 | 88 |
| Geometry | 5 | 48 | 86 | 13 | 6 | 0 | 161 |
| Calculus single-variable | 4 | 297 | 317 | 175 | 176 | 1 | 982 |
| Calculus multivariable | 13 | 279 | 244 | 80 | 37 | 0 | 654 |
| Linear Algebra | 2 | 172 | 232 | 53 | 26 | 0 | 498 |
| Number Theory | 2 | 33 | 7 | 1 | 1 | 2 | 46 |
| Financial Mathematics | 7 | 52 | 105 | 72 | 89 | 0 | 346 |
| Probability | 11 | 167 | 139 | 16 | 2 | 0 | 336 |
| Statistics | 44 | 151 | 158 | 46 | 0 | 0 | 401 |
| Complex Analysis | 0 | 40 | 7 | 2 | 1 | 1 | 51 |
| Differential Equations | 2 | 47 | 115 | 56 | 84 | 0 | 305 |
| Abstract Algebra | 0 | 8 | 16 | 9 | 1 | 7 | 42 |
| Grand Total | 145 | 1,855 | 1,751 | 657 | 588 | 12 | 5,062 |

Table 9: Answer Types and Definitions

| Answer Type | Definition |
|---|---|
| *Atomic Types* | |
| Numerical Value (NV) | Problems where the answer is a numerical value, including special values such as $\pi$, $e$, $\sqrt{7}$, $\sin \pi/8$, etc., represented in LaTeX. |
| Expression (EX) | Problems requiring an expression containing variables, e.g., $8x^2 + x + 1$, represented in LaTeX. |
| Equation (EQ) | Problems requiring an equation containing variables, e.g., $y = 2x + 1$ represented in LaTeX. |
| Interval (INT) | Problems where the answer is a range of values, e.g., $(-\infty, 2) \cup (3, \infty)$ represented as an interval in LaTeX. |
| True/False (TF) | Problems where the answer is either True or False, Yes or No, T or F, Y or N, etc. |
| MC with Single answer (MCS) | Multiple-Choice (MC) problems with only one correct option (e.g., one out of four, one out of five, etc.). The options can be captical letters (ABCD) or any other string according to the problems (independent or dependent, etc.). |
| MC with multiple answers (MCM) | Multiple-Choice (MC) problems with multiple correct options (e.g., two out of four, two out of five, two out of six, etc.). The options can be captical letters (ABCD) or any other string according to the problems (independent or dependent, etc.). |
| Open-Ended (OE) | Problems whose answers can be a term, name, or any other string that satisfies the description of the problem, for example, the name of the variable or function that occurs in the problem (which should be treated differently with EX). |
| *Compound Types* | |
| Ordered List (OL) | Problems where the answer is an ordered list, e.g. a coordinate ( $(1, 2, 3)$, $(2t, t^2)$, etc.). |
| Unordered List (UOL) | Problems where the answer is an unordered list, e.g., a set or multiple solutions for an equation. |

As mentioned in Appendix A.1, our problem generation programs will remain the main problem structure and knowledge points. One such example is illustrated in Figure 7 and the corresponding program is given in Listing 1. The dashed red box highlights the differences between randomized

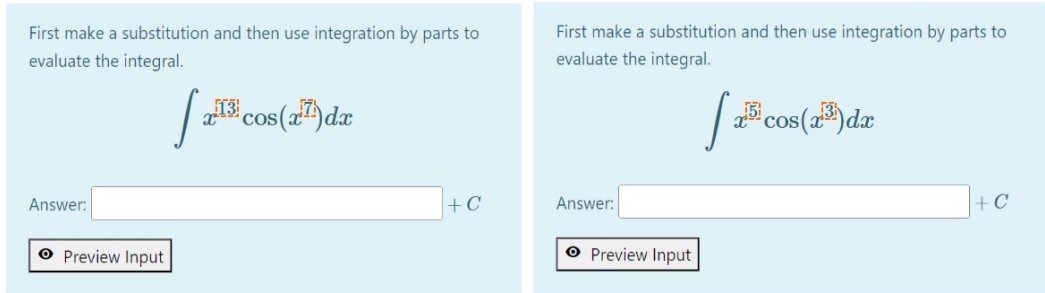

Figure 7: An Example of Our Online Homework Grading System Based on WebWork. The dashed red box highlights the differences between randomized versions of the problem.

versions of the problem. For this specific example, the second random variable is initialized through $a = random(2, 10)$. The relationship between the first and second variables is defined by $expnt = -1 + 2*a$. This meticulous setup maintains the consistency of the concepts, techniques, and solutions involved in different versions of each problem. Another example is demonstrated in Figure 8, where 'n' is initialized to a randomly chosen exponent from 2 to 10 in increments of 0.1.

All problems within WebWork are stored in the Problem Generation language and are presented in HTML format with JavaScript and external resources. This poses challenges for human interpretation and LLMs analysis, urging us to clean and re-format them in Latex (see Section 3.2).

```
$a=random(2, 10); # all-inclusive integers between 2 and 10
$expnt = -1+2*$a;
$ans = "1/$a*x^$a*sin(x^$a)+1/$a*cos(x^$a)"; #Right answer

TEXT(beginproblem());
BEGIN_TEXT
First make a substitution and then use integration by parts to evaluate the integral.
$BR
\[ \int x^{$expnt} \cos(x^$a) dx \]
$BR
Answer: \{ ans_rule(40)\}   \(+\) \(C\)
END_TEXT

#Compare the right answer to the input
ANS(fun_cmp($ans, mode=>'antider'));
```

Listing 1: Problem Generator Code for the Problem in Figure 7

## B.2   DEDUPLICATION

After converting and cleaning all the problems, we perform deduplication within each subject. More specifically, we adhere to the following steps: First, we transform each question into a vector with dimension 1536 using the embedding model `text-embedding-ada-002`, which is the most capable 2nd generation embedding model of OpenAI[7]. We then calculate pairwise cosine similarities using the embeddings in the previous step. Finally, a threshold is selected based on manual inspection within each subject, and problems that have a cosine similarity higher than that threshold with existing problems are excluded. The thresholds and the number of questions filtered out for different subjects are presented in Table 10. We use subject-agnostic thresholds and filter out 9,382 questions in total.

---

[7]The information can be found at https://openai.com/index/new-embedding-models-and-api-updates/

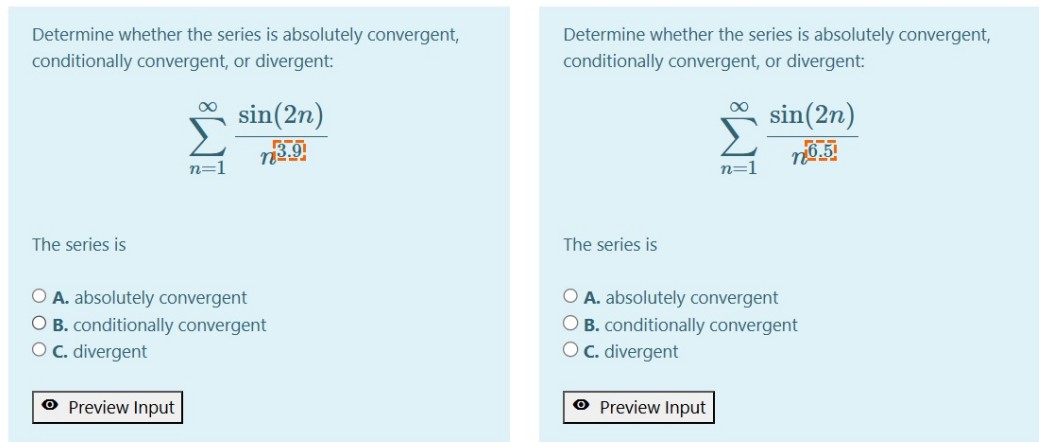

Figure 8: Another Example of Our Online Homework Grading System Based on WebWork.

Table 10: Thresholds of deduplication and the number of questions that filtered out in UGMathBench.

|  | Threshold | Before | After | # Filter Out |
|---|---|---|---|---|
| Arithmetic | 0.92 | 845 | 322 | 523 |
| Algebra | 0.86 | 4663 | 583 | 4080 |
| Set theory and logic | 0.96 | 94 | 69 | 25 |
| Trigonometry | 0.91 | 399 | 178 | 221 |
| Combinatorics | 0.89 | 140 | 88 | 52 |
| Geometry | 0.92 | 267 | 161 | 106 |
| Calculus single-variable | 0.90 | 3234 | 982 | 2252 |
| Calculus multivariable | 0.94 | 866 | 654 | 212 |
| Linear Algebra | 0.93 | 1235 | 498 | 737 |
| Number Theory | 0.92 | 68 | 46 | 22 |
| Financial Mathematics | 0.92 | 653 | 346 | 307 |
| Probability | 0.91 | 453 | 336 | 117 |
| Statistics | 0.92 | 599 | 401 | 198 |
| Complex Analysis | 0.95 | 129 | 51 | 78 |
| Differential Equations | 0.92 | 741 | 305 | 436 |
| Abstract Algebra | 0.94 | 58 | 42 | 16 |
| Grand Total | - | 14444 | 5,062 | 9382 |

## C  DETAILED EXPERIMENTAL SETUP

### C.1  EVALUATED LLMS

A variety of LLMs are covered in our evaluation, including closed-source commercial models and open-source models, general-purpose models and models dedicated for math problem solving. Closed-source LLMs are as follows:

- **o1-preview** (OpenAI, 2024b): An early preview of OpenAI's o1 model, designed to reason about hard problems using broad general knowledge about the world. We used o1-preview-2024-09-12 for our evaluation.
- **GPT-4o** (OpenAI, 2024a): GPT-4o is multimodal, and has the same high intelligence as GPT-4 Turbo but is much more efficient.
- **Claude-3-Opus** (Anthropic, 2024): Anthropic's most intelligent model, claimed to outperform its peers on most of the common evaluation benchmarks for AI systems.

We evaluated the following open-source general-purpose LLMs on our benchmark:

- **Llama-3-Instruct** (AI@Meta, 2024): LLaMA 3 Community License.

- **Mistral-7B-Instruct-v0.3** (Mistral, 2024a): Apache 2.0

- **Mistral-Nemo-Instruct-2407** (Mistral, 2024c): Apache 2.0

- **Mistral-Small-Instruct-2409** (Mistral, 2024d): MRL License.

- **Mistral-Large-Instruct-2407** (Mistral, 2024b): MRL License.

- **Qwen2-Instruct** (Yang et al., 2024a): Qwen2 series are developed with dedication to math and coding. We used 7B and 72B models. 7B models are licensed under Apache 2.0, while 72B models are under Tongyi Qianwen License.

- **Yi-1.5-Chat** (AI et al., 2024): Yi-1.5 delivers stronger performance in coding, math, reasoning, and instruction-following capability compared to its predecessor. We used 6B, 9B, 34B variants. Yi-1.5 series are licensed under Apache 2.0.

- **DeepSeek-V2-Lite-Chat** (DeepSeek-AI, 2024): model under Model License code under MIT License.

- **deepseek-moe-16b-chat** (Dai et al., 2024): model under Model License, code under MIT License.

The following specialized math LLMs are evaluated in our study:

- **DeepSeekMath-7B** (Shao et al., 2024): DeepSeekMath is initialized with DeepSeek-Coder-v1.5 7B and continues pre-training on math-related tokens. We tested both DeepSeekMath-7B-RL and DeepSeekMath-7B-Instruct variants. Models are under Model License while code is under MIT License.

- **Qwen2-Math** (Yang et al., 2024b): Qwen2-Math is a series of specialized math language models built upon the Qwen2 LLMs. We evaluated 7B and 72B variants. They are under the same license as Qwen2-Instruct series.

- **Mathstral-7B** (Mistral, 2023): Mathstral stands on the shoulders of Mistral 7B and specializes in STEM subjects. This model is published under Apache 2.0.

- **Numinamath-7B-CoT** (Beeching et al., 2024): This model is finetuned from DeepSeekMath-7B-base with two stages of supervised fine-tuning to solve math problems using chain of thought (CoT). It is licensed under Apache 2.0.

## C.2 EVALUATION PROMPTS

The evaluation prompts in our experiments are given in Table 11, where detailed answer type descriptions are given in Table 12. Following He et al. (2024a); Huang et al. (2024), these prompts are specially designed for different subjects and answer types for better evaluation. Note that, for chat models, we will apply chat template for better evaluation.

## D RESULTS ACROSS DIFFERENT SUBJECTS

The detailed results across different subjects are given in Table 13, 14, 15, 16, 17, 18, 19, 20, 21, 22, 23, 24, 25, 26, 27, and 28. From the results, we have the following observations:

- OpenAI-o1-mini achieves the best results across nearly all subjects, although GPT-4o sometimes excels in terms of RE.

- For open-source LLMs, Qwen-2-Math-72B-instruct achieves the best results in almost all subjects. However, Mistral-Large-instruct-2407 outperforms Qwen-2-Math-72B-instruct in Algebra.

- Some LLMs even achieve zero EAcc in certain subjects. For instance, Mistral-7B-Instruct get zero EAcc in Set Theory and Logic, and seven LLMs exhibit zero EAcc in Abstract Algebra.

- The variation in the reasoning gap differs significantly across subjects, providing more fine-grained information on how different LLMs perform across various domains.

Table 11: Evaluation prompts for problems with single answer or multiple answers. {problem} is the specific problem to evaluate. {subject} denotes the subject this problem belongs to and all subjects are given in Table 6. {answer_type_description} are specified in Table 12. {number_of_answers} stands for the number of answers in the problem evaluated.

| Evaluation Prompt for Single Answer |
|---|
| The following is an undergraduate-level mathematical problem in {subject}. You need to solve the problem by completing all placeholders [ANS]. |
| This problem involves only one placeholders [ANS] to be completed. The answer type is {answer_type_description}. |
| Problem:
{problem} |
| All mathematical formulas and symbols you output should be represented with LaTeX. Please end your response with: "The final answer is $\boxed{\text{ANSWER}}$, where ANSWER should be your final answer. |
| **Evaluation Prompt for Multiple Answers** |
| The following is an undergraduate-level mathematical problem in {subject}. You need to solve the problem by completing all placeholders [ANS]. |
| This problem involves {number_of_answers} placeholders [ANS] to be completed. Their answer types are, in order, {answer_type_description}. |
| Problem:
{problem} |
| All mathematical formulas and symbols you output should be represented with LaTeX. Please end your response with: "The final answer is $\boxed{\text{ANSWER}}$, where ANSWER should be the sequence of your final answers, separated by commas. |

Table 12: Descriptions of answer types included in evaluation prompts, where {options} is the specific options from the multiple choice question evaluated.

| Answer Type | Answer Type Description |
|---|---|
| NV | a numerical value without units |
| EX | an expression |
| EQ | an equation |
| INT | a range interval |
| TF | either True or False |
| MCS | one option for a multiple choice question with options {options} |
| MCM | more than one option concatenated without space or commas of a multiple choice question with options {options}, for example: BD |
| OE | a word, phrase, term or string that satisfies the requirements of the problem |
| OL | an ordered list of answers surrounded by parentheses with any answer types, for example $(1, x^2, True)$, where "ordered list" means changing the order of elements results in different answers |
| UOL | an unordered list of answers surrounded by parentheses with any answer types, for example, $(1, x^2, True)$, where "unordered list" means changing the order of elements results in the same answer |

Table 13: **Results on Abstract Algebra** (all figures are in %). The best results within each column are **bolded** and the best results of open-source Chat LLMs within a similar parameter size group are underlined.

| Models | $Acc_1$ | $Acc_2$ | $Acc_3$ | AAcc | EAcc | $\Delta$ | RE |
|---|---|---|---|---|---|---|---|
| *Closed-source LLMs* | | | | | | | |
| OpenAI-o1-mini-2024-09-12 | **76.19** | **73.81** | **71.42** | **73.81** | **57.14** | 16.67 | **29.17** |
| GPT-4o-2024-08-06 | 42.86 | 50.00 | 52.38 | 48.41 | 28.57 | 19.84 | 69.44 |
| GPT-4o-mini-2024-07-18 | 26.19 | 35.71 | 38.09 | 33.33 | 14.29 | 19.04 | 133.2 |
| Claude-3-Opus-20240229 | 23.81 | 38.10 | 26.19 | 29.37 | 11.90 | 17.47 | 146.8 |
| *Open-source Chat LLMs* | | | | | | | |
| Yi-1.5-6B-Chat | 4.76 | 2.38 | 9.52 | 5.56 | 0.00 | 5.56 | $\infty$ |
| Mistral-7B-Instruct | 2.38 | 2.38 | 0.00 | 1.59 | 0.00 | **1.59** | $\infty$ |
| Qwen2-7B-Instruct | 4.76 | 16.67 | 19.05 | 13.49 | 4.76 | 8.73 | 183.4 |
| LLaMA3-8B-Instruct | 0.00 | 7.14 | 9.52 | 5.56 | 0.00 | 5.56 | $\infty$ |
| Yi-1.5-9B-Chat | 7.14 | 14.29 | 11.90 | 11.11 | 0.00 | 11.11 | $\infty$ |
| Mistral-Nemo-Instruct-2407 | 0.00 | 9.52 | 4.76 | 4.76 | 0.00 | 4.76 | $\infty$ |
| DeepSeek-MOE-16B-Chat | 0.00 | 2.38 | 2.38 | 1.59 | 0.00 | **1.59** | $\infty$ |
| DeepSeek-V2-Lite-Chat | 0.00 | 2.38 | 2.38 | 1.59 | 0.00 | **1.59** | $\infty$ |
| Mistral-Small-Instruct-2409 | 9.52 | 19.05 | 21.43 | 16.67 | 7.14 | 9.53 | 133.5 |
| Yi-1.5-34B-Chat | 19.05 | 9.52 | 9.52 | 12.70 | 0.00 | 12.70 | $\infty$ |
| LLaMA3-70B-Instruct | 19.05 | 16.67 | 19.04 | 18.25 | 4.76 | 13.49 | 283.4 |
| Qwen2-72B-Instruct | 28.57 | 47.62 | 30.95 | 35.71 | 14.29 | 21.42 | 149.9 |
| Mistral-Large-Instruct-2407 | 35.71 | 35.71 | 45.24 | 38.89 | 21.43 | 17.46 | 81.47 |
| *Specialized Mathematical LLMs* | | | | | | | |
| DeepSeek-Math-7B-Instruct | 7.14 | 9.52 | 4.76 | 7.14 | 0.00 | 7.14 | $\infty$ |
| DeepSeek-Math-7B-RL | 4.76 | 19.05 | 11.90 | 11.90 | 2.38 | 9.52 | 400.0 |
| NuminaMath-7B-CoT | 7.14 | 16.67 | 7.14 | 10.32 | 0.00 | 10.32 | $\infty$ |
| Mathstral-7B-v0.1 | 9.52 | 4.76 | 7.14 | 7.14 | 0.00 | 7.14 | $\infty$ |
| Qwen2-Math-7B-Instruct | 23.81 | 33.33 | 28.57 | 28.57 | 14.29 | 14.28 | 99.93 |
| Qwen2-Math-72B-Instruct | 45.24 | 40.48 | 42.86 | 42.86 | 26.19 | 16.67 | 63.65 |

# E  ERROR ANALYSIS

We perform error analysis in Section 5.4, and here we showcase several examples of various error types in Table 29, 30, and 31.

The distribution of the relative error for OpenAI-o1-mini numerical values is illustrated in Figure 9. We excluded numerical answers identical to the ground truth, as their logarithmic relative error would be negative infinity.

# F  RESULTS ACROSS DIFFERENT TOPICS

The performances of differerent LLMs on 20 topics are shown in Figure 10, 11, 12 and 13. We observe that different LLMs exhibit varying performance patterns across these topics, and even models within the same family show differences in their rankings.

# G  REFINEMENT RESULTS

Progressive-Hint Prompting (PHP) (Zheng et al., 2023) is a technique designed to enhance automatic, iterative interactions with LLMs. PHP uses previously generated answers as hints to progressively guide users toward the correct solutions. In our experiments, we employ the zero-shot manner to ensure a fair comparison with the primary experiments in Table 4, which helps to assess the impact of refinement on the performance of LLMs in solving undergraduate-level mathematical problems. We have set the maximum number of interaction rounds to five. To save cost, we only experiment with GPT-4o for closed-source LLMs. The results are presented in Table 32. While PHP can improve AAcc and EAcc in most cases, the improvements are not substantial. There remains considerable potential for enhancing the mathematical reasoning abilities of LLMs in solving undergraduate-level

Table 14: **Results on Algebra** (all figures are in %). The best results within each column are **bolded** and the best results of open-source Chat LLMs within a similar parameter size group are underlined.

| Models | Acc$_1$ | Acc$_2$ | Acc$_3$ | AAcc | EAcc | $\Delta$ | RE |
|---|---|---|---|---|---|---|---|
| *Closed-source LLMs* | | | | | | | |
| OpenAI-o1-mini-2024-09-12 | **76.16** | **75.47** | **74.44** | **75.36** | **66.72** | 8.64 | 12.95 |
| GPT-4o-2024-08-06 | 70.50 | 71.53 | 72.04 | 71.36 | 64.67 | **6.69** | **10.34** |
| GPT-4o-mini-2024-07-18 | 67.41 | 67.24 | 66.38 | 67.01 | 59.52 | 7.49 | 12.58 |
| Claude-3-Opus-20240229 | 63.29 | 66.38 | 63.81 | 64.49 | 54.20 | 10.29 | 18.99 |
| *Open-source Chat LLMs* | | | | | | | |
| Yi-1.5-6B-Chat | 38.77 | 37.22 | 40.48 | 38.82 | 25.21 | 13.61 | 53.99 |
| Mistral-7B-Instruct | 19.73 | 21.96 | 19.55 | 20.41 | 11.32 | 9.09 | 80.30 |
| Qwen2-7B-Instruct | 51.63 | 53.00 | 54.03 | 52.89 | 41.85 | 11.04 | 26.38 |
| LLaMA3-8B-Instruct | 30.53 | 33.28 | 30.19 | 31.33 | 18.35 | 12.98 | 70.74 |
| Yi-1.5-9B-Chat | 46.83 | 48.89 | 48.37 | 48.03 | 33.79 | 14.24 | 42.14 |
| Mistral-Nemo-Instruct-2407 | 39.79 | 43.05 | 39.79 | 40.88 | 28.82 | 12.06 | 41.85 |
| DeepSeek-MOE-16B-Chat | 12.01 | 12.35 | 10.98 | 11.78 | 4.97 | 6.81 | 137.0 |
| DeepSeek-V2-Lite-Chat | 23.33 | 24.01 | 24.53 | 23.96 | 11.32 | 12.64 | 111.7 |
| Mistral-Small-Instruct-2409 | 58.32 | 57.46 | 56.95 | 57.58 | 47.86 | 9.72 | 20.31 |
| Yi-1.5-34B-Chat | 53.34 | 53.00 | 52.66 | 53.00 | 40.31 | 12.69 | 31.48 |
| LLaMA3-70B-Instruct | 50.09 | 51.80 | 51.11 | 51.00 | 40.48 | 10.52 | 25.99 |
| Qwen2-72B-Instruct | 63.81 | 63.46 | 64.49 | 63.92 | 54.72 | 9.20 | 16.81 |
| Mistral-Large-Instruct-2407 | 68.44 | 68.10 | 67.92 | 68.15 | 59.86 | 8.29 | 13.85 |
| *Specialized Mathematical LLMs* | | | | | | | |
| DeepSeek-Math-7B-Instruct | 40.31 | 38.77 | 38.25 | 39.11 | 26.07 | 13.04 | 50.02 |
| DeepSeek-Math-7B-RL | 44.43 | 47.51 | 44.94 | 45.63 | 34.31 | 11.32 | 32.99 |
| NuminaMath-7B-CoT | 45.80 | 45.45 | 46.14 | 45.80 | 34.65 | 11.15 | 32.18 |
| Mathstral-7B-v0.1 | 45.97 | 43.91 | 43.57 | 44.48 | 32.25 | 12.23 | 37.92 |
| Qwen2-Math-7B-Instruct | 56.09 | 57.46 | 57.63 | 57.06 | 47.86 | 9.20 | 19.22 |
| Qwen2-Math-72B-Instruct | 67.58 | 67.75 | 67.24 | 67.52 | 58.83 | 8.69 | 14.77 |

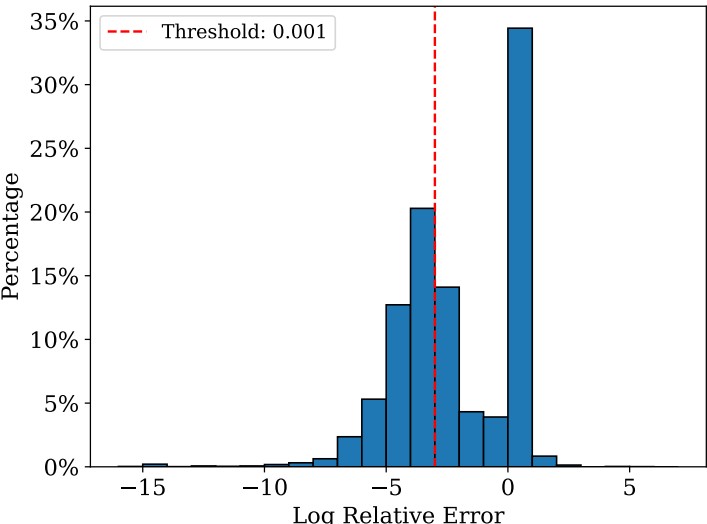

Figure 9: Relative Error of Numeric Answers excluding ones that are identical to the ground truth. Red dotted line indicates the tolerance threshold when we evaluate model answers.

mathematics. The results for PHP across different subjects for GPT-4o (see Table 33) indicate that the impact of PHP is subject-agnostic.

Table 15: **Main Results on Arithmetic** (all figures are in %). The best results within each column are **bolded** and the best results of open-source Chat LLMs within a similar parameter size group are underlined.

| Models | $Acc_1$ | $Acc_2$ | $Acc_3$ | AAcc | EAcc | $\Delta$ | RE |
|---|---|---|---|---|---|---|---|
| *Closed-source LLMs* | | | | | | | |
| OpenAI-o1-mini-2024-09-12 | **93.48** | **92.86** | **94.10** | **93.48** | **90.37** | **3.11** | **3.44** |
| GPT-4o-2024-08-06 | 91.93 | 91.30 | 92.24 | 91.82 | 87.27 | 4.55 | 5.21 |
| GPT-4o-mini-2024-07-18 | 89.44 | 87.27 | 90.06 | 88.92 | 83.23 | 5.69 | 6.84 |
| Claude-3-Opus-20240229 | 84.78 | 85.71 | 86.96 | 85.82 | 78.26 | 7.56 | 9.66 |
| *Open-source Chat LLMs* | | | | | | | |
| Yi-1.5-6B-Chat | 63.98 | 64.91 | 67.70 | 65.53 | 50.00 | 15.53 | 31.06 |
| Mistral-7B-Instruct | 30.43 | 34.78 | 33.54 | 32.92 | 15.22 | 17.70 | 116.3 |
| Qwen2-7B-Instruct | 77.64 | 78.26 | 78.57 | 78.16 | 68.94 | 9.22 | 13.37 |
| LLaMA3-8B-Instruct | 51.24 | 53.42 | 54.97 | 53.21 | 33.54 | 19.67 | 58.65 |
| Yi-1.5-9B-Chat | 73.60 | 73.60 | 70.81 | 72.67 | 59.32 | 13.35 | 22.51 |
| Mistral-Nemo-Instruct-2407 | 69.88 | 70.19 | 72.67 | 70.91 | 55.28 | 15.63 | 28.27 |
| DeepSeek-MOE-16B-Chat | 26.09 | 30.43 | 31.68 | 29.40 | 12.42 | 16.98 | 136.7 |
| DeepSeek-V2-Lite-Chat | 50.31 | 53.42 | 49.07 | 50.93 | 32.92 | 18.01 | 54.71 |
| Mistral-Small-Instruct-2409 | 82.61 | 84.78 | 81.68 | 83.02 | 73.29 | 9.73 | 13.28 |
| Yi-1.5-34B-Chat | 75.47 | 74.84 | 75.47 | 75.26 | 62.42 | 12.84 | 20.57 |
| LLaMA3-70B-Instruct | 79.81 | 82.61 | 83.85 | 82.09 | 72.36 | 9.73 | 13.45 |
| Qwen2-72B-Instruct | 88.51 | 88.20 | 89.44 | 88.72 | 82.61 | 6.11 | 7.40 |
| Mistral-Large-Instruct-2407 | 90.37 | 89.13 | 90.06 | 89.86 | 82.61 | 7.25 | 8.78 |
| *Specialized Mathematical LLMs* | | | | | | | |
| DeepSeek-Math-7B-Instruct | 62.73 | 68.63 | 65.22 | 65.53 | 49.69 | 15.84 | 31.88 |
| DeepSeek-Math-7B-RL | 74.53 | 76.71 | 76.71 | 75.98 | 67.08 | 8.90 | 13.27 |
| NuminaMath-7B-CoT | 74.22 | 75.78 | 77.33 | 75.78 | 62.42 | 13.36 | 21.40 |
| Mathstral-7B-v0.1 | 71.74 | 72.67 | 72.36 | 72.26 | 59.63 | 12.63 | 21.18 |
| Qwen2-Math-7B-Instruct | 88.51 | 84.78 | 86.65 | 86.65 | 78.88 | 7.77 | 9.85 |
| Qwen2-Math-72B-Instruct | 90.68 | 90.68 | 90.99 | 90.79 | 86.96 | 3.83 | 4.40 |

# H    REASONING GAP AND TEST SET CONTAMINATION

Given the limited number of examples in the test set for one version (in terms of SFT), it is necessary to mix the test set with general mathematical SFT data. For our experiments, we adopt MetaMathQA (Yu et al., 2023), a high-quality SFT dataset for math word problems, which includes 395,000 training examples. We use Llama-3-8B as our base model and set the maximum output token length to 4096. Following Tong et al. (2024), we set the learning rate to 5e-5, use a warmup ratio of 0.03, adopt cosine decay, and train the model for one epoch. Training one model takes approximately three hours on four A100 GPUs. The results are presented in Table 5. We set the proportion of the test set (from one version) for SFT to 5%, 10%, 15% and, 20% and see how the reasoning gap varies. As the proportion of the test set included in the training data increases, the reasoning gap ($\Delta$))also becomes more pronounced. This study serves as an initial investigation into test set contamination during the SFT stage. It is important to note that contamination at the pre-training stage is also a significant area of interest (Razeghi et al., 2022; Jiang et al., 2024).

Table 16: **Main Results on Calculus - multivariable** (all figures are in %). The best results within each column are **bolded** and the best results of open-source Chat LLMs within a similar parameter size group are underlined.

| Models | Acc$_1$ | Acc$_2$ | Acc$_3$ | AAcc | EAcc | $\Delta$ | RE |
|---|---|---|---|---|---|---|---|
| *Closed-source LLMs* | | | | | | | |
| OpenAI-o1-mini-2024-09-12 | **59.33** | **60.09** | **59.94** | **59.79** | **48.32** | 11.47 | **23.74** |
| GPT-4o-2024-08-06 | 50.00 | 49.69 | 49.24 | 49.64 | 38.23 | 11.41 | 29.85 |
| GPT-4o-mini-2024-07-18 | 42.81 | 44.80 | 43.27 | 43.63 | 32.11 | 11.52 | 35.88 |
| Claude-3-Opus-20240229 | 35.93 | 39.14 | 35.93 | 37.00 | 24.01 | 12.99 | 54.10 |
| *Open-source Chat LLMs* | | | | | | | |
| Yi-1.5-6B-Chat | 20.34 | 22.48 | 20.03 | 20.95 | 10.40 | 10.55 | 101.4 |
| Mistral-7B-Instruct | 5.05 | 3.98 | 4.59 | 4.54 | 1.22 | 3.32 | 272.1 |
| Qwen2-7B-Instruct | 25.38 | 25.23 | 26.15 | 25.59 | 16.06 | 9.53 | 59.34 |
| LLaMA3-8B-Instruct | 6.27 | 7.80 | 7.03 | 7.03 | 2.45 | 4.58 | 186.9 |
| Yi-1.5-9B-Chat | 28.75 | 28.75 | 29.51 | 29.00 | 16.67 | 12.33 | 73.97 |
| Mistral-Nemo-Instruct-2407 | 13.61 | 12.39 | 10.55 | 12.18 | 5.81 | 6.37 | 109.6 |
| DeepSeek-MOE-16B-Chat | 1.68 | 0.46 | 1.22 | 1.12 | 0.00 | **1.12** | $\infty$ |
| DeepSeek-V2-Lite-Chat | 5.05 | 4.89 | 3.52 | 4.49 | 1.38 | 3.11 | 225.4 |
| Mistral-Small-Instruct-2409 | 27.98 | 29.51 | 27.68 | 28.39 | 17.28 | 11.11 | 64.29 |
| Yi-1.5-34B-Chat | 26.45 | 27.98 | 26.15 | 26.86 | 14.22 | 12.64 | 88.89 |
| LLaMA3-70B-Instruct | 20.79 | 23.39 | 20.95 | 21.71 | 12.23 | 9.48 | 77.51 |
| Qwen2-72B-Instruct | 35.02 | 37.92 | 35.32 | 36.09 | 23.55 | 12.54 | 53.25 |
| Mistral-Large-Instruct-2407 | 46.94 | 49.08 | 47.71 | 47.91 | 36.70 | 11.21 | 30.54 |
| *Specialized Mathematical LLMs* | | | | | | | |
| DeepSeek-Math-7B-Instruct | 11.16 | 12.54 | 12.23 | 11.98 | 5.35 | 6.63 | 123.9 |
| DeepSeek-Math-7B-RL | 18.81 | 17.58 | 16.06 | 17.48 | 8.56 | 8.92 | 104.2 |
| NuminaMath-7B-CoT | 20.49 | 20.80 | 22.02 | 21.10 | 11.47 | 9.63 | 83.96 |
| Mathstral-7B-v0.1 | 16.36 | 17.89 | 17.89 | 17.38 | 9.63 | 7.75 | 80.48 |
| Qwen2-Math-7B-Instruct | 35.78 | 36.09 | 33.33 | 35.07 | 24.77 | 10.30 | 41.58 |
| Qwen2-Math-72B-Instruct | 48.47 | 49.39 | 51.22 | 49.69 | 39.45 | 10.24 | 25.96 |

Table 17: **Main Results on Calculus - single variable** (all figures are in %). The best results within each column are **bolded** and the best results of open-source Chat LLMs within a similar parameter size group are underlined.

| Models | Acc$_1$ | Acc$_2$ | Acc$_3$ | AAcc | EAcc | $\Delta$ | RE |
|---|---|---|---|---|---|---|---|
| *Closed-source LLMs* | | | | | | | |
| OpenAI-o1-mini-2024-09-12 | **72.81** | **73.32** | **72.61** | **72.91** | **59.37** | 13.54 | 22.81 |
| GPT-4o-2024-08-06 | 63.65 | 65.48 | 65.48 | 64.87 | 53.05 | 11.82 | **22.28** |
| GPT-4o-mini-2024-07-18 | 54.68 | 57.43 | 55.91 | 56.01 | 42.26 | 13.75 | 32.53 |
| Claude-3-Opus-20240229 | 47.96 | 51.32 | 50.61 | 49.97 | 34.73 | 15.24 | 43.88 |
| *Open-source Chat LLMs* | | | | | | | |
| Yi-1.5-6B-Chat | 27.39 | 31.57 | 30.96 | 29.97 | 16.80 | 13.17 | 78.39 |
| Mistral-7B-Instruct | 8.86 | 10.29 | 8.96 | 9.37 | 4.38 | 4.99 | 113.9 |
| Qwen2-7B-Instruct | 39.51 | 41.96 | 39.41 | 40.29 | 27.70 | 12.59 | 45.45 |
| LLaMA3-8B-Instruct | 13.14 | 13.95 | 13.44 | 13.51 | 7.03 | 6.48 | 92.18 |
| Yi-1.5-9B-Chat | 37.27 | 37.88 | 38.19 | 37.78 | 23.42 | 14.36 | 61.32 |
| Mistral-Nemo-Instruct-2407 | 21.28 | 23.22 | 22.20 | 22.23 | 12.32 | 9.91 | 80.44 |
| DeepSeek-MOE-16B-Chat | 4.58 | 4.48 | 4.48 | 4.51 | 1.22 | 3.29 | 269.67 |
| DeepSeek-V2-Lite-Chat | 11.30 | 12.53 | 12.02 | 11.95 | 4.38 | 7.57 | 172.83 |
| Mistral-Small-Instruct-2409 | 39.41 | 40.73 | 39.51 | 39.88 | 26.37 | 13.51 | 51.23 |
| Yi-1.5-34B-Chat | 39.41 | 41.34 | 39.21 | 39.99 | 24.95 | 15.04 | 60.28 |
| LLaMA3-70B-Instruct | 32.89 | 36.46 | 34.21 | 34.52 | 22.81 | 11.71 | 51.34 |
| Qwen2-72B-Instruct | 47.86 | 50.20 | 48.88 | 48.98 | 33.50 | 15.48 | 46.21 |
| Mistral-Large-Instruct-2407 | 57.33 | 59.17 | 59.57 | 58.69 | 45.93 | 12.76 | 27.78 |
| *Specialized Mathematical LLMs* | | | | | | | |
| DeepSeek-Math-7B-Instruct | 23.83 | 27.39 | 23.83 | 25.02 | 13.75 | 11.27 | 81.96 |
| DeepSeek-Math-7B-RL | 30.04 | 30.65 | 30.65 | 30.45 | 20.06 | 10.39 | 51.79 |
| NuminaMath-7B-CoT | 32.08 | 32.89 | 32.79 | 32.59 | 20.16 | 12.43 | 61.66 |
| Mathstral-7B-v0.1 | 27.70 | 27.19 | 29.53 | 28.14 | 16.29 | 11.85 | 72.74 |
| Qwen2-Math-7B-Instruct | 46.03 | 49.49 | 49.49 | 48.34 | 35.74 | 12.60 | 35.25 |
| Qwen2-Math-72B-Instruct | 61.71 | 61.81 | 60.08 | 61.20 | 49.29 | 11.91 | 24.16 |

Table 18: **Main Results on Combinatorics** (all figures are in %). Models are classified into three categories according to their purpose and origin. The best results within each column are **bolded** and the best results of open-source Chat LLMs within a similar parameter size group are underlined.

| Models | Acc$_1$ | Acc$_2$ | Acc$_3$ | AAcc | EAcc | $\Delta$ | RE |
|---|---|---|---|---|---|---|---|
| *Closed-source LLMs* | | | | | | | |
| OpenAI-o1-mini-2024-09-12 | **90.91** | **92.05** | **95.45** | **92.80** | **88.64** | **4.16** | **4.69** |
| GPT-4o-2024-08-06 | 77.27 | 77.27 | 78.41 | 77.65 | 61.36 | 16.29 | 26.55 |
| GPT-4o-mini-2024-07-18 | 70.45 | 69.32 | 73.86 | 71.21 | 56.82 | 14.39 | 25.33 |
| Claude-3-Opus-20240229 | 56.82 | 65.91 | 68.18 | 63.64 | 50.00 | 13.64 | 27.28 |
| *Open-source Chat LLMs* | | | | | | | |
| Yi-1.5-6B-Chat | 25.00 | 34.09 | 29.55 | 29.55 | 18.18 | 11.37 | 62.54 |
| Mistral-7B-Instruct | 10.23 | 12.50 | 11.36 | 11.36 | 2.27 | 9.09 | 400.4 |
| Qwen2-7B-Instruct | 39.77 | 45.45 | 44.32 | 43.18 | 28.41 | 14.77 | 51.99 |
| LLaMA3-8B-Instruct | 22.73 | 23.86 | 23.86 | 23.48 | 14.77 | 8.71 | 58.97 |
| Yi-1.5-9B-Chat | 32.95 | 43.18 | 44.32 | 40.15 | 25.00 | 15.15 | 60.60 |
| Mistral-Nemo-Instruct-2407 | 26.14 | 30.68 | 32.95 | 29.92 | 18.18 | 11.74 | 64.58 |
| DeepSeek-MOE-16B-Chat | 6.82 | 11.36 | 7.95 | 8.71 | 2.27 | 6.44 | 283.7 |
| DeepSeek-V2-Lite-Chat | 17.05 | 22.73 | 12.50 | 17.42 | 6.82 | 10.60 | 155.43 |
| Mistral-Small-Instruct-2409 | 46.59 | 59.09 | 50.00 | 51.89 | 38.64 | 13.25 | 34.29 |
| Yi-1.5-34B-Chat | 37.50 | 48.86 | 52.27 | 46.21 | 27.27 | 18.94 | 69.45 |
| LLaMA3-70B-Instruct | 47.73 | 52.27 | 45.45 | 48.48 | 37.50 | 10.98 | 29.28 |
| Qwen2-72B-Instruct | 60.23 | 63.64 | 60.23 | 61.36 | 47.73 | 13.63 | 28.56 |
| Mistral-Large-Instruct-2407 | 69.32 | 72.73 | 71.59 | 71.21 | 57.95 | 13.26 | 22.88 |
| *Specialized Mathematical LLMs* | | | | | | | |
| DeepSeek-Math-7B-Instruct | 26.14 | 37.50 | 27.27 | 30.30 | 17.05 | 13.25 | 77.71 |
| DeepSeek-Math-7B-RL | 30.68 | 42.05 | 39.77 | 37.50 | 26.14 | 11.36 | 43.46 |
| NuminaMath-7B-CoT | 30.68 | 35.23 | 37.50 | 34.47 | 23.86 | 10.61 | 44.47 |
| Mathstral-7B-v0.1 | 34.09 | 38.64 | 35.23 | 35.98 | 26.14 | 9.84 | 37.64 |
| Qwen2-Math-7B-Instruct | 62.50 | 62.50 | 64.77 | 63.26 | 47.73 | 15.53 | 32.54 |
| Qwen2-Math-72B-Instruct | 76.14 | 76.14 | 75.00 | 75.76 | 62.50 | 13.26 | 21.22 |

Table 19: **Main Results on Complex analysis** (all figures are in %). The best results within each column are **bolded** and the best results of open-source Chat LLMs within a similar parameter size group are underlined.

| Models | Acc$_1$ | Acc$_2$ | Acc$_3$ | AAcc | EAcc | $\Delta$ | RE |
|---|---|---|---|---|---|---|---|
| *Closed-source LLMs* | | | | | | | |
| OpenAI-o1-mini-2024-09-12 | **74.51** | **74.51** | **76.47** | **75.16** | **64.71** | 10.45 | 16.15 |
| GPT-4o-2024-08-06 | 66.67 | 70.59 | 70.59 | 69.28 | 58.82 | 10.46 | 17.78 |
| GPT-4o-mini-2024-07-18 | 62.75 | 62.75 | 62.75 | 62.75 | 50.98 | 11.77 | 23.09 |
| Claude-3-Opus-20240229 | 47.06 | 49.02 | 50.98 | 49.02 | 37.25 | 11.77 | 31.60 |
| *Open-source Chat LLMs* | | | | | | | |
| Yi-1.5-6B-Chat | 35.29 | 37.25 | 31.37 | 34.64 | 23.53 | 11.11 | 47.22 |
| Mistral-7B-Instruct | 11.76 | 11.76 | 9.80 | 11.11 | 5.88 | 5.23 | 88.95 |
| Qwen2-7B-Instruct | 45.10 | 50.98 | 43.14 | 46.41 | 35.29 | 11.12 | 31.51 |
| LLaMA3-8B-Instruct | 19.61 | 17.65 | 17.65 | 18.30 | 13.73 | 4.57 | 33.28 |
| Yi-1.5-9B-Chat | 39.22 | 39.22 | 39.22 | 39.22 | 27.45 | 11.77 | 42.88 |
| Mistral-Nemo-Instruct-2407 | 29.41 | 25.49 | 25.49 | 26.80 | 19.61 | 7.19 | 36.66 |
| DeepSeek-MOE-16B-Chat | 1.96 | 3.92 | 1.96 | 2.61 | 0.00 | 2.61 | $\infty$ |
| DeepSeek-V2-Lite-Chat | 17.65 | 13.73 | 15.69 | 15.69 | 7.84 | 7.85 | 100.1 |
| Mistral-Small-Instruct-2409 | 47.06 | 49.02 | 43.14 | 46.41 | 37.25 | 9.16 | 24.59 |
| Yi-1.5-34B-Chat | 47.06 | 45.10 | 39.22 | 43.79 | 25.49 | 18.30 | 71.79 |
| LLaMA3-70B-Instruct | 43.14 | 33.33 | 37.25 | 37.91 | 25.49 | 12.42 | 48.72 |
| Qwen2-72B-Instruct | 52.94 | 62.75 | 62.75 | 59.48 | 47.06 | 12.42 | 26.39 |
| Mistral-Large-Instruct-2407 | 64.71 | 64.71 | 60.78 | 63.40 | 52.94 | 10.46 | 19.76 |
| *Specialized Mathematical LLMs* | | | | | | | |
| DeepSeek-Math-7B-Instruct | 35.29 | 29.41 | 27.45 | 30.72 | 23.53 | 7.19 | 30.56 |
| DeepSeek-Math-7B-RL | 35.29 | 41.18 | 47.06 | 41.18 | 27.45 | 13.73 | 50.02 |
| NuminaMath-7B-CoT | 33.33 | 35.29 | 37.25 | 35.29 | 21.57 | 13.72 | 63.61 |
| Mathstral-7B-v0.1 | 37.25 | 29.41 | 31.37 | 32.68 | 15.69 | 16.99 | 108.3 |
| Qwen2-Math-7B-Instruct | 56.86 | 49.02 | 52.94 | 52.94 | 41.18 | 11.76 | 28.56 |
| Qwen2-Math-72B-Instruct | 70.59 | 72.55 | 72.55 | 71.90 | 64.71 | 7.19 | **11.11** |

Table 20: **Main Results on Differential equations** (all figures are in %). Models are classified into three categories according to their purpose and origin. The best results within each column are **bolded** and the best results of open-source Chat LLMs within a similar parameter size group are underlined.

| Models | Acc$_1$ | Acc$_2$ | Acc$_3$ | AAcc | EAcc | $\Delta$ | RE |
|---|---|---|---|---|---|---|---|
| *Closed-source LLMs* | | | | | | | |
| OpenAI-o1-mini-2024-09-12 | **29.84** | **29.18** | **29.51** | **29.51** | **22.62** | 6.89 | 30.46 |
| GPT-4o-2024-08-06 | 23.93 | 24.59 | 25.57 | 24.70 | 19.34 | 5.36 | **27.71** |
| GPT-4o-mini-2024-07-18 | 20.00 | 22.30 | 19.67 | 20.66 | 16.07 | 4.59 | 28.56 |
| Claude-3-Opus-20240229 | 18.69 | 18.03 | 20.33 | 19.02 | 13.11 | 5.91 | 45.08 |
| *Open-source Chat LLMs* | | | | | | | |
| Yi-1.5-6B-Chat | 6.23 | 9.51 | 4.59 | 6.78 | 2.95 | 3.83 | 129.8 |
| Mistral-7B-Instruct | 3.28 | 3.93 | 4.59 | 3.93 | 1.64 | 2.29 | 139.6 |
| Qwen2-7B-Instruct | 11.80 | 12.13 | 11.48 | 11.80 | 7.54 | 4.26 | 56.50 |
| LLaMA3-8B-Instruct | 4.26 | 5.90 | 5.57 | 5.25 | 2.62 | 2.63 | 100.4 |
| Yi-1.5-9B-Chat | 11.80 | 13.44 | 12.13 | 12.46 | 6.23 | 6.23 | 100.0 |
| Mistral-Nemo-Instruct-2407 | 8.52 | 7.87 | 9.18 | 8.52 | 4.26 | 4.26 | 100.0 |
| DeepSeek-MOE-16B-Chat | 0.66 | 1.31 | 0.00 | 0.66 | 0.00 | **0.66** | $\infty$ |
| DeepSeek-V2-Lite-Chat | 2.62 | 2.62 | 1.97 | 2.40 | 0.33 | 2.07 | 627.3 |
| Mistral-Small-Instruct-2409 | 15.74 | 15.74 | 14.43 | 15.30 | 9.84 | 5.46 | 55.49 |
| Yi-1.5-34B-Chat | 12.13 | 15.08 | 12.13 | 13.11 | 7.21 | 5.90 | 81.83 |
| LLaMA3-70B-Instruct | 12.79 | 13.44 | 12.13 | 12.79 | 7.54 | 5.25 | 69.63 |
| Qwen2-72B-Instruct | 19.34 | 20.00 | 18.36 | 19.23 | 13.77 | 5.46 | 39.65 |
| Mistral-Large-Instruct-2407 | 22.95 | 25.90 | 24.92 | 24.59 | 18.69 | 5.90 | 31.57 |
| *Specialized Mathematical LLMs* | | | | | | | |
| DeepSeek-Math-7B-Instruct | 6.89 | 7.54 | 5.90 | 6.78 | 1.97 | 4.81 | 244.2 |
| DeepSeek-Math-7B-RL | 8.52 | 10.82 | 9.84 | 9.73 | 5.25 | 4.48 | 85.33 |
| NuminaMath-7B-CoT | 10.82 | 10.16 | 7.87 | 9.62 | 4.26 | 5.36 | 125.8 |
| Mathstral-7B-v0.1 | 10.49 | 10.49 | 9.51 | 10.16 | 4.92 | 5.24 | 106.5 |
| Qwen2-Math-7B-Instruct | 15.41 | 18.69 | 18.03 | 17.38 | 12.13 | 5.25 | 43.28 |
| Qwen2-Math-72B-Instruct | 22.95 | 22.95 | 22.95 | 22.95 | 17.38 | 5.57 | 32.05 |

Table 21: **Main Results on Financial mathematics** (all figures are in %). Models are classified into three categories according to their purpose and origin. The best results within each column are **bolded** and the best results of open-source Chat LLMs within a similar parameter size group are underlined.

| Models | Acc$_1$ | Acc$_2$ | Acc$_3$ | AAcc | EAcc | $\Delta$ | RE |
|---|---|---|---|---|---|---|---|
| *Closed-source LLMs* | | | | | | | |
| OpenAI-o1-mini-2024-09-12 | **46.53** | **49.13** | **51.16** | **48.94** | **28.32** | 20.62 | 72.81 |
| GPT-4o-2024-08-06 | 28.61 | 29.77 | 29.77 | 29.38 | 18.21 | 11.17 | 61.34 |
| GPT-4o-mini-2024-07-18 | 16.19 | 17.63 | 19.65 | 17.82 | 9.54 | 8.28 | 86.79 |
| Claude-3-Opus-20240229 | 19.65 | 21.39 | 20.52 | 20.52 | 10.98 | 9.54 | 86.89 |
| *Open-source Chat LLMs* | | | | | | | |
| Yi-1.5-6B-Chat | 5.49 | 6.94 | 6.94 | 6.45 | 3.18 | 3.27 | 102.8 |
| Mistral-7B-Instruct | 1.16 | 3.47 | 2.02 | 2.22 | 0.87 | 1.35 | 155.2 |
| Qwen2-7B-Instruct | 8.67 | 12.43 | 8.96 | 10.02 | 3.76 | 6.26 | 166.5 |
| LLaMA3-8B-Instruct | 2.60 | 2.02 | 3.18 | 2.60 | 1.16 | 1.44 | 124.1 |
| Yi-1.5-9B-Chat | 9.25 | 14.45 | 12.43 | 12.04 | 4.05 | 7.99 | 197.2 |
| Mistral-Nemo-Instruct-2407 | 4.62 | 5.49 | 6.07 | 5.39 | 1.73 | 3.66 | 211.6 |
| DeepSeek-MOE-16B-Chat | 0.58 | 0.87 | 1.16 | 0.87 | 0.00 | **0.87** | $\infty$ |
| DeepSeek-V2-Lite-Chat | 3.76 | 3.76 | 3.76 | 3.76 | 1.45 | 2.31 | 159.3 |
| Mistral-Small-Instruct-2409 | 12.43 | 11.85 | 13.01 | 12.43 | 5.78 | 6.65 | 115.1 |
| Yi-1.5-34B-Chat | 11.85 | 14.45 | 13.58 | 13.29 | 5.20 | 8.09 | 155.6 |
| LLaMA3-70B-Instruct | 8.09 | 10.12 | 9.25 | 9.15 | 4.62 | 4.53 | 98.05 |
| Qwen2-72B-Instruct | 15.03 | 18.50 | 16.47 | 16.67 | 8.96 | 7.71 | 86.05 |
| Mistral-Large-Instruct-2407 | 22.54 | 26.59 | 21.97 | 23.70 | 15.32 | 8.38 | **54.70** |
| *Specialized Mathematical LLMs* | | | | | | | |
| DeepSeek-Math-7B-Instruct | 4.91 | 6.07 | 4.62 | 5.20 | 2.31 | 2.89 | 125.1 |
| DeepSeek-Math-7B-RL | 5.49 | 7.51 | 5.78 | 6.26 | 2.02 | 4.24 | 209.9 |
| NuminaMath-7B-CoT | 8.67 | 9.25 | 9.25 | 9.06 | 3.76 | 5.30 | 141.0 |
| Mathstral-7B-v0.1 | 4.91 | 9.83 | 7.23 | 7.32 | 3.18 | 4.14 | 130.2 |
| Qwen2-Math-7B-Instruct | 10.69 | 17.05 | 15.32 | 14.35 | 5.49 | 8.86 | 161.38 |
| Qwen2-Math-72B-Instruct | 28.90 | 26.88 | 29.19 | 28.32 | 16.47 | 11.85 | 71.95 |

Table 22: **Main Results on Geometry** (all figures are in %). Models are classified into three categories according to their purpose and origin. The best results within each column are **bolded** and the best results of open-source Chat LLMs within a similar parameter size group are underlined.

| Models | $Acc_1$ | $Acc_2$ | $Acc_3$ | AAcc | EAcc | $\Delta$ | RE |
|---|---|---|---|---|---|---|---|
| *Closed-source LLMs* | | | | | | | |
| OpenAI-o1-mini-2024-09-12 | **71.43** | **67.08** | **71.43** | **69.98** | **58.39** | 11.59 | 19.85 |
| GPT-4o-2024-08-06 | 64.60 | 63.98 | 63.98 | 64.18 | 51.55 | 12.63 | 24.50 |
| GPT-4o-mini-2024-07-18 | 60.87 | 55.90 | 59.63 | 58.80 | 44.10 | 14.70 | 33.33 |
| Claude-3-Opus-20240229 | 57.76 | 49.69 | 49.07 | 52.17 | 37.89 | 14.28 | 37.69 |
| *Open-source Chat LLMs* | | | | | | | |
| Yi-1.5-6B-Chat | 32.92 | 27.33 | 30.43 | 30.23 | 18.01 | 12.22 | 67.85 |
| Mistral-7B-Instruct | 8.70 | 8.70 | 6.83 | 8.07 | 3.11 | 4.96 | 159.5 |
| Qwen2-7B-Instruct | 50.31 | 39.75 | 42.86 | 44.31 | 29.19 | 15.12 | 51.80 |
| LLaMA3-8B-Instruct | 24.84 | 19.88 | 19.88 | 21.53 | 9.94 | 11.59 | 116.6 |
| Yi-1.5-9B-Chat | 38.51 | 37.89 | 36.02 | 37.47 | 22.36 | 15.11 | 67.58 |
| Mistral-Nemo-Instruct-2407 | 31.68 | 27.95 | 28.57 | 29.40 | 16.77 | 12.63 | 75.31 |
| DeepSeek-MOE-16B-Chat | 4.97 | 3.73 | 6.21 | 4.97 | 1.86 | **3.11** | 167.2 |
| DeepSeek-V2-Lite-Chat | 11.18 | 14.29 | 14.91 | 13.46 | 4.97 | 8.49 | 170.8 |
| Mistral-Small-Instruct-2409 | 44.72 | 47.21 | 46.58 | 46.17 | 32.92 | 13.25 | 40.25 |
| Yi-1.5-34B-Chat | 43.48 | 37.89 | 44.72 | 42.03 | 27.33 | 14.70 | 53.79 |
| LLaMA3-70B-Instruct | 39.13 | 38.51 | 39.75 | 39.13 | 25.47 | 13.66 | 53.63 |
| Qwen2-72B-Instruct | 59.01 | 49.69 | 53.42 | 54.04 | 40.99 | 13.05 | 31.84 |
| Mistral-Large-Instruct-2407 | 61.49 | 57.76 | 60.25 | 59.83 | 50.31 | 9.52 | **18.92** |
| *Specialized Mathematical LLMs* | | | | | | | |
| DeepSeek-Math-7B-Instruct | 29.19 | 24.22 | 24.22 | 25.88 | 14.29 | 11.59 | 81.11 |
| DeepSeek-Math-7B-RL | 34.78 | 29.19 | 29.81 | 31.26 | 19.25 | 12.01 | 62.39 |
| NuminaMath-7B-CoT | 33.54 | 31.68 | 28.57 | 31.26 | 19.25 | 12.01 | 62.39 |
| Mathstral-7B-v0.1 | 39.13 | 32.92 | 37.89 | 36.65 | 24.22 | 12.43 | 51.32 |
| Qwen2-Math-7B-Instruct | 52.80 | 42.86 | 45.96 | 47.20 | 34.16 | 13.04 | 38.17 |
| Qwen2-Math-72B-Instruct | 67.08 | 61.49 | 64.60 | 64.39 | 53.42 | 10.97 | 20.54 |

Table 23: **Main Results on Linear algebra** (all figures are in %). Models are classified into three categories according to their purpose and origin. The best results within each column are **bolded** and the best results of open-source Chat LLMs within a similar parameter size group are underlined.

| Models | $Acc_1$ | $Acc_2$ | $Acc_3$ | AAcc | EAcc | $\Delta$ | RE |
|---|---|---|---|---|---|---|---|
| *Closed-source LLMs* | | | | | | | |
| OpenAI-o1-mini-2024-09-12 | **74.90** | **74.90** | **75.90** | **75.23** | **65.86** | 9.37 | **14.23** |
| GPT-4o-2024-08-06 | 63.86 | 61.65 | 61.65 | 62.38 | 51.61 | 10.77 | 20.87 |
| GPT-4o-mini-2024-07-18 | 57.63 | 57.03 | 56.22 | 56.96 | 43.37 | 13.59 | 31.34 |
| Claude-3-Opus-20240229 | 51.20 | 48.80 | 49.00 | 49.67 | 37.15 | 12.52 | 33.70 |
| *Open-source Chat LLMs* | | | | | | | |
| Yi-1.5-6B-Chat | 23.69 | 23.69 | 21.49 | 22.96 | 12.45 | 10.51 | 84.42 |
| Mistral-7B-Instruct | 8.03 | 7.43 | 7.23 | 7.56 | 2.61 | 4.95 | 189.7 |
| Qwen2-7B-Instruct | 36.35 | 34.54 | 32.73 | 34.54 | 22.29 | 12.25 | 54.96 |
| LLaMA3-8B-Instruct | 13.25 | 12.45 | 12.65 | 12.78 | 6.43 | 6.35 | 98.76 |
| Yi-1.5-9B-Chat | 34.14 | 31.93 | 31.93 | 32.66 | 18.47 | 14.19 | 76.83 |
| Mistral-Nemo-Instruct-2407 | 23.69 | 20.48 | 23.90 | 22.69 | 11.24 | 11.45 | 101.9 |
| DeepSeek-MOE-16B-Chat | 2.01 | 2.41 | 3.61 | 2.68 | 0.80 | **1.88** | 235.0 |
| DeepSeek-V2-Lite-Chat | 9.24 | 8.23 | 7.43 | 8.30 | 1.81 | 6.49 | 358.6 |
| Mistral-Small-Instruct-2409 | 43.17 | 38.35 | 41.57 | 41.03 | 28.51 | 12.52 | 43.91 |
| Yi-1.5-34B-Chat | 35.34 | 31.93 | 34.94 | 34.07 | 20.08 | 13.99 | 69.67 |
| LLaMA3-70B-Instruct | 34.34 | 30.72 | 29.32 | 31.46 | 19.28 | 12.18 | 63.17 |
| Qwen2-72B-Instruct | 52.81 | 51.00 | 49.80 | 51.20 | 38.15 | 13.05 | 34.21 |
| Mistral-Large-Instruct-2407 | 62.25 | 59.04 | 58.43 | 59.91 | 48.59 | 11.32 | 23.30 |
| *Specialized Mathematical LLMs* | | | | | | | |
| DeepSeek-Math-7B-Instruct | 20.08 | 21.08 | 18.07 | 19.75 | 9.44 | 10.31 | 109.2 |
| DeepSeek-Math-7B-RL | 26.91 | 26.51 | 23.09 | 25.50 | 15.46 | 10.04 | 64.94 |
| NuminaMath-7B-CoT | 25.70 | 25.50 | 27.11 | 26.10 | 14.66 | 11.44 | 78.04 |
| Mathstral-7B-v0.1 | 29.92 | 23.90 | 24.70 | 26.17 | 13.86 | 12.31 | 88.82 |
| Qwen2-Math-7B-Instruct | 46.39 | 43.17 | 43.57 | 44.38 | 32.73 | 11.65 | 35.59 |
| Qwen2-Math-72B-Instruct | 61.45 | 59.24 | 62.85 | 61.18 | 47.59 | 13.59 | 28.56 |

Table 24: **Main Results on Number theory** (all figures are in %). Models are classified into three categories according to their purpose and origin. The best results within each column are **bolded** and the best results of open-source Chat LLMs within a similar parameter size group are underlined.

| Models | Acc$_1$ | Acc$_2$ | Acc$_3$ | AAcc | EAcc | $\Delta$ | RE |
|---|---|---|---|---|---|---|---|
| *Closed-source LLMs* | | | | | | | |
| OpenAI-o1-mini-2024-09-12 | **80.43** | **76.09** | **73.91** | **76.81** | **58.70** | 18.11 | 30.85 |
| GPT-4o-2024-08-06 | 60.87 | 63.04 | 67.39 | 63.77 | 56.52 | 7.25 | **12.83** |
| GPT-4o-mini-2024-07-18 | 60.87 | 58.70 | 58.70 | 59.42 | 52.17 | 7.25 | 13.90 |
| Claude-3-Opus-20240229 | 47.83 | 54.35 | 56.52 | 52.90 | 39.13 | 13.77 | 35.19 |
| *Open-source Chat LLMs* | | | | | | | |
| Yi-1.5-6B-Chat | 26.09 | 26.09 | 19.57 | 23.91 | 13.04 | 10.87 | 83.36 |
| Mistral-7B-Instruct | 6.52 | 6.52 | 8.70 | 7.25 | 2.17 | 5.08 | 234.1 |
| Qwen2-7B-Instruct | 34.78 | 39.13 | 41.30 | 38.41 | 23.91 | 14.50 | 60.64 |
| LLaMA3-8B-Instruct | 19.57 | 21.74 | 15.22 | 18.84 | 8.70 | 10.14 | 116.6 |
| Yi-1.5-9B-Chat | 34.78 | 32.61 | 34.78 | 34.06 | 23.91 | 10.15 | 42.45 |
| Mistral-Nemo-Instruct-2407 | 17.39 | 28.26 | 28.26 | 24.64 | 8.70 | 15.94 | 183.2 |
| DeepSeek-MOE-16B-Chat | 6.52 | 6.52 | 2.17 | 5.07 | 2.17 | **2.90** | 133.6 |
| DeepSeek-V2-Lite-Chat | 15.22 | 13.04 | 15.22 | 14.49 | 4.35 | 10.14 | 233.1 |
| Mistral-Small-Instruct-2409 | 45.65 | 34.78 | 41.30 | 40.58 | 30.43 | 10.15 | 33.36 |
| Yi-1.5-34B-Chat | 36.96 | 34.78 | 47.83 | 39.86 | 26.09 | 13.77 | 52.78 |
| LLaMA3-70B-Instruct | 30.43 | 23.91 | 36.96 | 30.43 | 15.22 | 15.21 | 99.93 |
| Qwen2-72B-Instruct | 47.83 | 58.70 | 43.48 | 50.00 | 36.96 | 13.04 | 35.28 |
| Mistral-Large-Instruct-2407 | 58.70 | 58.70 | 58.70 | 58.70 | 45.65 | 13.05 | 28.59 |
| *Specialized Mathematical LLMs* | | | | | | | |
| DeepSeek-Math-7B-Instruct | 23.91 | 28.26 | 30.43 | 27.54 | 15.22 | 12.32 | 80.95 |
| DeepSeek-Math-7B-RL | 21.74 | 26.09 | 26.09 | 24.64 | 13.04 | 11.60 | 88.96 |
| NuminaMath-7B-CoT | 17.39 | 32.61 | 32.61 | 27.54 | 13.04 | 14.50 | 111.2 |
| Mathstral-7B-v0.1 | 28.26 | 30.43 | 30.43 | 29.71 | 17.39 | 12.32 | 70.85 |
| Qwen2-Math-7B-Instruct | 39.13 | 43.48 | 43.48 | 42.03 | 30.43 | 11.60 | 38.12 |
| Qwen2-Math-72B-Instruct | 60.87 | 54.35 | 63.04 | 59.42 | 50.00 | 9.42 | 18.84 |

Table 25: **Main Results on Probability** (all figures are in %). Models are classified into three categories according to their purpose and origin. The best results within each column are **bolded** and the best results of open-source Chat LLMs within a similar parameter size group are underlined.

| Models | Acc$_1$ | Acc$_2$ | Acc$_3$ | AAcc | EAcc | $\Delta$ | RE |
|---|---|---|---|---|---|---|---|
| *Closed-source LLMs* | | | | | | | |
| OpenAI-o1-mini-2024-09-12 | **66.07** | **68.45** | **68.15** | **67.56** | 51.49 | 16.07 | 31.21 |
| GPT-4o-2024-08-06 | 66.07 | 66.07 | 65.77 | 65.97 | **54.76** | 11.21 | **20.47** |
| GPT-4o-mini-2024-07-18 | 50.89 | 56.85 | 52.38 | 53.37 | 39.88 | 13.49 | 33.83 |
| Claude-3-Opus-20240229 | 56.55 | 59.23 | 59.82 | 58.53 | 44.94 | 13.59 | 30.24 |
| *Open-source Chat LLMs* | | | | | | | |
| Yi-1.5-6B-Chat | 20.24 | 22.62 | 21.43 | 21.43 | 10.42 | 11.01 | 105.7 |
| Mistral-7B-Instruct | 10.12 | 11.01 | 10.42 | 10.52 | 3.57 | 6.95 | 194.7 |
| Qwen2-7B-Instruct | 31.85 | 39.58 | 33.63 | 35.02 | 20.54 | 14.48 | 70.50 |
| LLaMA3-8B-Instruct | 16.37 | 16.96 | 15.48 | 16.27 | 9.23 | 7.04 | 76.27 |
| Yi-1.5-9B-Chat | 28.27 | 30.06 | 31.85 | 30.06 | 13.69 | 16.37 | 119.6 |
| Mistral-Nemo-Instruct-2407 | 23.81 | 26.79 | 25.00 | 25.20 | 15.18 | 10.02 | 66.01 |
| DeepSeek-MOE-16B-Chat | 5.65 | 4.76 | 4.17 | 4.86 | 1.19 | **3.67** | 308.4 |
| DeepSeek-V2-Lite-Chat | 9.82 | 11.61 | 10.42 | 10.62 | 3.27 | 7.35 | 224.8 |
| Mistral-Small-Instruct-2409 | 39.58 | 43.15 | 43.45 | 42.06 | 27.68 | 14.38 | 51.95 |
| Yi-1.5-34B-Chat | 37.20 | 41.67 | 38.39 | 39.09 | 23.51 | 15.58 | 66.27 |
| LLaMA3-70B-Instruct | 35.12 | 36.31 | 34.23 | 35.22 | 22.02 | 13.20 | 59.95 |
| Qwen2-72B-Instruct | 48.51 | 48.51 | 44.05 | 47.02 | 33.04 | 13.98 | 42.31 |
| Mistral-Large-Instruct-2407 | 59.82 | 60.12 | 58.33 | 59.42 | 45.54 | 13.88 | 30.48 |
| *Specialized Mathematical LLMs* | | | | | | | |
| DeepSeek-Math-7B-Instruct | 21.73 | 22.92 | 21.43 | 22.02 | 10.71 | 11.31 | 105.6 |
| DeepSeek-Math-7B-RL | 24.40 | 25.89 | 26.19 | 25.50 | 14.88 | 10.62 | 71.37 |
| NuminaMath-7B-CoT | 25.60 | 26.49 | 25.30 | 25.79 | 13.39 | 12.40 | 92.61 |
| Mathstral-7B-v0.1 | 27.38 | 29.46 | 25.30 | 27.38 | 14.88 | 12.50 | 84.01 |
| Qwen2-Math-7B-Instruct | 36.61 | 44.35 | 39.88 | 40.28 | 25.89 | 14.39 | 55.58 |
| Qwen2-Math-72B-Instruct | 58.33 | 62.20 | 59.52 | 60.02 | 44.05 | 15.97 | 36.25 |

Table 26: **Main Results on Set theory and logic** (all figures are in %). Models are classified into three categories according to their purpose and origin. The best results within each column are **bolded** and the best results of open-source Chat LLMs within a similar parameter size group are underlined.

| Models | Acc$_1$ | Acc$_2$ | Acc$_3$ | AAcc | EAcc | $\Delta$ | RE |
|---|---|---|---|---|---|---|---|
| *Closed-source LLMs* | | | | | | | |
| OpenAI-o1-mini-2024-09-12 | **78.26** | 73.91 | **79.71** | **77.29** | **66.67** | 10.62 | **15.93** |
| GPT-4o-2024-08-06 | 75.36 | **76.81** | 68.12 | 73.43 | 59.42 | 14.01 | 23.58 |
| GPT-4o-mini-2024-07-18 | 59.42 | 57.97 | 59.42 | 58.94 | 40.58 | 18.36 | 45.24 |
| Claude-3-Opus-20240229 | 56.52 | 60.87 | 62.32 | 59.90 | 43.48 | 16.42 | 37.76 |
| *Open-source Chat LLMs* | | | | | | | |
| Yi-1.5-6B-Chat | 14.49 | 14.49 | 18.84 | 15.94 | 7.25 | 8.69 | 119.9 |
| Mistral-7B-Instruct | 2.90 | 5.80 | 5.80 | 4.83 | 0.00 | 4.83 | $\infty$ |
| Qwen2-7B-Instruct | 20.29 | 36.23 | 24.64 | 27.05 | 13.04 | 14.01 | 107.4 |
| LLaMA3-8B-Instruct | 8.70 | 15.94 | 8.70 | 11.11 | 5.80 | 5.31 | 91.55 |
| Yi-1.5-9B-Chat | 24.64 | 18.84 | 26.09 | 23.19 | 5.80 | 17.39 | 299.8 |
| Mistral-Nemo-Instruct-2407 | 24.64 | 33.33 | 26.09 | 28.02 | 18.84 | 9.18 | 48.73 |
| DeepSeek-MOE-16B-Chat | 2.90 | 1.45 | 1.45 | 1.93 | 0.00 | **1.93** | $\infty$ |
| DeepSeek-V2-Lite-Chat | 4.35 | 7.25 | 5.80 | 5.80 | 1.45 | 4.35 | 300.0 |
| Mistral-Small-Instruct-2409 | 47.83 | 44.93 | 40.58 | 44.44 | 30.43 | 14.01 | 46.04 |
| Yi-1.5-34B-Chat | 30.43 | 40.58 | 43.48 | 38.16 | 21.74 | 16.42 | 75.53 |
| LLaMA3-70B-Instruct | 28.99 | 26.09 | 24.64 | 26.57 | 15.94 | 10.63 | 66.69 |
| Qwen2-72B-Instruct | 52.17 | 55.07 | 49.28 | 52.17 | 36.23 | 15.94 | 44.00 |
| Mistral-Large-Instruct-2407 | 59.42 | 60.87 | 57.97 | 59.42 | 44.93 | 14.49 | 32.25 |
| *Specialized Mathematical LLMs* | | | | | | | |
| DeepSeek-Math-7B-Instruct | 21.74 | 18.84 | 13.04 | 17.87 | 8.70 | 9.17 | 105.4 |
| DeepSeek-Math-7B-RL | 23.19 | 23.19 | 24.64 | 23.67 | 10.14 | 13.53 | 133.4 |
| NuminaMath-7B-CoT | 17.39 | 26.09 | 18.84 | 20.77 | 8.70 | 12.07 | 138.7 |
| Mathstral-7B-v0.1 | 23.19 | 26.09 | 20.29 | 23.19 | 14.49 | 8.70 | 60.04 |
| Qwen2-Math-7B-Instruct | 31.88 | 39.13 | 27.54 | 32.85 | 18.84 | 14.01 | 74.36 |
| Qwen2-Math-72B-Instruct | 60.87 | 56.52 | 55.07 | 57.49 | 40.58 | 16.91 | 41.67 |

Table 27: **Main Results on Statistics** (all figures are in %). Models are classified into three categories according to their purpose and origin. The best results within each column are **bolded** and the best results of open-source Chat LLMs within a similar parameter size group are underlined.

| Models | Acc$_1$ | Acc$_2$ | Acc$_3$ | AAcc | EAcc | $\Delta$ | RE |
|---|---|---|---|---|---|---|---|
| *Closed-source LLMs* | | | | | | | |
| OpenAI-o1-mini-2024-09-12 | **64.09** | **62.34** | **61.85** | **62.76** | **49.88** | 12.88 | 25.82 |
| GPT-4o-2024-08-06 | 59.10 | 62.09 | 60.10 | 60.43 | 49.38 | 11.05 | **22.38** |
| GPT-4o-mini-2024-07-18 | 43.14 | 46.38 | 48.63 | 46.05 | 32.92 | 13.13 | 39.88 |
| Claude-3-Opus-20240229 | 52.12 | 53.12 | 50.37 | 51.87 | 37.16 | 14.71 | 39.59 |
| *Open-source Chat LLMs* | | | | | | | |
| Yi-1.5-6B-Chat | 15.96 | 16.96 | 21.45 | 18.12 | 6.23 | 11.89 | 190.9 |
| Mistral-7B-Instruct | 11.97 | 11.72 | 11.47 | 11.72 | 2.99 | 8.73 | 291.0 |
| Qwen2-7B-Instruct | 26.68 | 27.93 | 29.18 | 27.93 | 13.97 | 13.96 | 99.93 |
| LLaMA3-8B-Instruct | 10.72 | 13.47 | 16.21 | 13.47 | 5.99 | 7.48 | 124.9 |
| Yi-1.5-9B-Chat | 26.43 | 23.94 | 31.67 | 27.35 | 13.97 | 13.38 | 95.78 |
| Mistral-Nemo-Instruct-2407 | 22.94 | 27.18 | 24.94 | 25.02 | 14.71 | 10.31 | 70.09 |
| DeepSeek-MOE-16B-Chat | 3.24 | 3.49 | 4.74 | 3.82 | 1.00 | **2.82** | 282.0 |
| DeepSeek-V2-Lite-Chat | 8.98 | 8.48 | 9.23 | 8.89 | 2.74 | 6.15 | 224.5 |
| Mistral-Small-Instruct-2409 | 37.66 | 35.91 | 35.66 | 36.41 | 23.69 | 12.72 | 53.69 |
| Yi-1.5-34B-Chat | 32.92 | 37.41 | 37.91 | 36.08 | 21.45 | 14.63 | 68.21 |
| LLaMA3-70B-Instruct | 22.44 | 20.70 | 21.45 | 21.53 | 11.97 | 9.56 | 79.87 |
| Qwen2-72B-Instruct | 43.14 | 44.39 | 44.89 | 44.14 | 31.42 | 12.72 | 40.48 |
| Mistral-Large-Instruct-2407 | 54.36 | 50.12 | 53.62 | 52.70 | 39.90 | 12.80 | 32.08 |
| *Specialized Mathematical LLMs* | | | | | | | |
| DeepSeek-Math-7B-Instruct | 18.95 | 16.96 | 18.95 | 18.29 | 6.73 | 11.56 | 171.8 |
| DeepSeek-Math-7B-RL | 21.70 | 24.44 | 22.69 | 22.94 | 9.23 | 13.71 | 148.5 |
| NuminaMath-7B-CoT | 17.71 | 18.20 | 18.95 | 18.29 | 6.48 | 11.81 | 182.3 |
| Mathstral-7B-v0.1 | 19.70 | 22.69 | 21.70 | 21.36 | 9.73 | 11.63 | 119.5 |
| Qwen2-Math-7B-Instruct | 31.42 | 31.17 | 37.66 | 33.42 | 17.71 | 15.71 | 88.71 |
| Qwen2-Math-72B-Instruct | 47.38 | 50.87 | 47.63 | 48.63 | 33.92 | 14.71 | 43.37 |

Table 28: **Main Results on Trigonometry** (all figures are in %). Models are classified into three categories according to their purpose and origin. The best results within each column are **bolded** and the best results of open-source Chat LLMs within a similar parameter size group are underlined.

| Models | Acc$_1$ | Acc$_2$ | Acc$_3$ | AAcc | EAcc | $\Delta$ | RE |
|---|---|---|---|---|---|---|---|
| *Closed-source LLMs* | | | | | | | |
| OpenAI-o1-mini-2024-09-12 | **75.84** | **74.16** | **69.10** | **73.03** | **56.74** | 16.29 | 28.71 |
| GPT-4o-2024-08-06 | 67.98 | 74.16 | 66.29 | 69.48 | 51.69 | 17.79 | 34.42 |
| GPT-4o-mini-2024-07-18 | 52.25 | 58.99 | 55.06 | 55.43 | 37.64 | 17.79 | 47.26 |
| Claude-3-Opus-20240229 | 53.93 | 52.81 | 53.37 | 53.37 | 37.64 | 15.73 | 41.79 |
| *Open-source Chat LLMs* | | | | | | | |
| Yi-1.5-6B-Chat | 24.16 | 24.72 | 20.79 | 23.22 | 11.24 | 11.98 | 106.6 |
| Mistral-7B-Instruct | 6.74 | 7.87 | 6.18 | 6.93 | 1.69 | 5.24 | 310.0 |
| Qwen2-7B-Instruct | 36.52 | 41.01 | 42.13 | 39.89 | 25.84 | 14.05 | 54.37 |
| LLaMA3-8B-Instruct | 14.61 | 12.92 | 13.48 | 13.67 | 4.49 | 9.18 | 204.5 |
| Yi-1.5-9B-Chat | 32.02 | 30.34 | 32.02 | 31.46 | 15.73 | 15.73 | 100.0 |
| Mistral-Nemo-Instruct-2407 | 23.03 | 23.60 | 24.72 | 23.78 | 11.80 | 11.98 | 101.5 |
| DeepSeek-MOE-16B-Chat | 3.93 | 3.93 | 4.49 | 4.12 | 0.00 | **4.12** | $\infty$ |
| DeepSeek-V2-Lite-Chat | 10.67 | 15.73 | 11.80 | 12.73 | 3.37 | 9.36 | 277.7 |
| Mistral-Small-Instruct-2409 | 38.76 | 41.01 | 45.51 | 41.76 | 27.53 | 14.23 | 51.69 |
| Yi-1.5-34B-Chat | 44.38 | 39.33 | 37.64 | 40.45 | 25.28 | 15.17 | 60.01 |
| LLaMA3-70B-Instruct | 33.71 | 36.52 | 34.83 | 35.02 | 23.03 | 11.99 | 52.06 |
| Qwen2-72B-Instruct | 52.25 | 50.56 | 50.00 | 50.94 | 35.39 | 15.55 | 43.94 |
| Mistral-Large-Instruct-2407 | 65.73 | 64.61 | 66.85 | 65.73 | 50.00 | 15.73 | 31.46 |
| *Specialized Mathematical LLMs* | | | | | | | |
| DeepSeek-Math-7B-Instruct | 26.40 | 28.09 | 29.78 | 28.09 | 11.24 | 16.85 | 149.9 |
| DeepSeek-Math-7B-RL | 32.02 | 33.71 | 36.52 | 34.08 | 20.22 | 13.86 | 68.55 |
| NuminaMath-7B-CoT | 33.71 | 34.83 | 30.34 | 32.96 | 17.42 | 15.54 | 89.21 |
| Mathstral-7B-v0.1 | 30.34 | 31.46 | 34.27 | 32.02 | 18.54 | 13.48 | 72.71 |
| Qwen2-Math-7B-Instruct | 53.93 | 50.00 | 51.69 | 51.87 | 38.76 | 13.11 | 33.82 |
| Qwen2-Math-72B-Instruct | 62.92 | 65.17 | 63.48 | 63.86 | 50.00 | 13.86 | **27.72** |

Table 29: An example of Linear algebra with expression calculation error.

---

**Problem**: You have boxes of five types. They are characterized by their length $L$, their width $W$, their height $H$, their age $A$, and their price $P$, as given in the following table:

| type | $L$ | $W$ | $H$ | $A$ | $P$ |
|------|-----|-----|-----|-----|-----|
| $I$ | 1 | 1 | 1 | 1 | 1 |
| $II$ | 1 | 1 | 2 | 3 | 2 |
| $III$ | 1 | 2 | 3 | 2 | 2 |
| $IV$ | 1 | 2 | 4 | 1 | 2 |
| $V$ | 1 | 2 | 5 | 1 | 3 |

So for example, boxes of type IV measure 1 by 2 by 4 feet, are a year old, and cost \$2 each. Suppose the sum of the lengths of your boxes is 25, the sum of their widths is 40, the sum of their heights is 75, the sum of their ages is 43, and the total price of those boxes is 51. You have [ANS] boxes of type I, [ANS] boxes of type II, [ANS] boxes of type III, [ANS] boxes of type IV, [ANS] boxes of type V.

---

**o1-mini Completion**: To solve the system of equations based on the given constraints, we define the variables as follows:

$$\begin{cases} x_1 = \text{Number of Type I boxes} \\ x_2 = \text{Number of Type II boxes} \\ x_3 = \text{Number of Type III boxes} \\ x_4 = \text{Number of Type IV boxes} \\ x_5 = \text{Number of Type V boxes} \end{cases}$$

Using the provided data, we set up the following equations:

1. $x_1 + x_2 + x_3 + x_4 + x_5 = 25$
2. $x_1 + x_2 + 2x_3 + 2x_4 + 2x_5 = 40$
3. $x_1 + 2x_2 + 3x_3 + 4x_4 + 5x_5 = 75$
4. $x_1 + 3x_2 + 2x_3 + x_4 + x_5 = 43$
5. $x_1 + 2x_2 + 2x_3 + 2x_4 + 3x_5 = 51$

By systematically solving these equations, we find the values of the variables:

$$x_5 = 2.5$$
$$x_1 = x_5 - 1 = 1.5$$
$$x_2 = 11 - x_5 = 8.5$$
$$x_3 = x_5 + 1 = 3.5$$
$$x_4 = 14 - 2x_5 = 9$$

[It skipped detailed steps and obtained incorrect results in equation manipulation]

**The final answers are $\boxed{1.5, 8.5, 3.5, 9, 2.5}$ **

---

**Ground-Truth Answer**: 4, 6, 6, 4, 5

---

**Model Output**: 1.5, 8.5, 3.5, 9, 2.5

---

Table 30: An example of Financial mathematics with numeric calculation error.

---

**Problem**: How long will it take for $3100 compounded semiannually at an annual rate of $1\%$ to amount to $7000? [ANS] years

---

**o1-mini Completion**: To determine how long it will take for $3100 to grow to $7000 with semiannual compounding at an annual rate of $1\%$, we can use the compound interest formula:

$$A = P\left(1 + \frac{r}{n}\right)^{nt}$$

Where: - $A = 7000$ (the final amount) - $P = 3100$ (the principal amount) - $r = 0.01$ (the annual interest rate) - $n = 2$ (the number of compounding periods per year) - $t$ is the time in years
Plugging in the known values:

$$7000 = 3100\left(1 + \frac{0.01}{2}\right)^{2t}$$

$$\frac{7000}{3100} = (1.005)^{2t}$$

$$2.258064516 = (1.005)^{2t}$$

Taking the natural logarithm of both sides:

$$\ln(2.258064516) = 2t \cdot \ln(1.005)$$

$$t = \frac{\ln(2.258064516)}{2 \cdot \ln(1.005)}$$

Calculating the values:

$$\ln(2.258064516) \approx 0.8154244$$
$$\text{[should be 0.8145080376]}$$
$$\ln(1.005) \approx 0.00498752$$
$$\text{[should be 0.00498754141]}$$
$$t \approx \frac{0.8154244}{2 \cdot 0.00498752} \approx 81.56$$
$$\left[\text{With the wrong parameters, should be } \frac{0.8154244}{2 \cdot 0.00498752} = 81.74647921\right]$$
$$\left[\text{With the right parameters, should be } \frac{0.8145080376}{2 \cdot 0.00498754141} = 81.6542615\right]$$

The final answer is $\boxed{81.56}$.

---

**Ground-Truth Answer**: 81.6542614995239

---

**Model Output**: 81.56

---

Table 31: An example of Financial mathematics with flawed reasoning.

**Problem**: If you make a deposit into a bank account, at what interest rate (compounded weekly) should you invest if you would like to double your investment in 78 weeks? [ANS] % (Note: Your answer should be accurate to two decimal places)

**o1-mini Completion**: To determine the interest rate $r$ (compounded weekly) required to double your investment in 78 weeks, we can use the compound interest formula:

$$FV = PV \times \left(1 + \frac{r}{100}\right)^n$$

$$\left[\text{here should be } \left(1 + \frac{r}{52}\right)^n \text{ for 52 weeks a year}\right]$$

Where: - $FV$ is the future value - $PV$ is the present value - $r$ is the weekly interest rate (in percentage) - $n$ is the number of weeks
Given that you want to double your investment:

$$2 = \left(1 + \frac{r}{100}\right)^{78}$$

To solve for $r$, take the natural logarithm of both sides:

$$\ln(2) = 78 \times \ln\left(1 + \frac{r}{100}\right)$$

$$\ln\left(1 + \frac{r}{100}\right) = \frac{\ln(2)}{78}$$

$$1 + \frac{r}{100} = e^{\frac{\ln(2)}{78}}$$

$$\frac{r}{100} = e^{\frac{\ln(2)}{78}} - 1$$

$$r = \left(e^{\frac{\ln(2)}{78}} - 1\right) \times 100$$

Calculating the above:

$$r \approx \left(e^{\frac{0.693147}{78}} - 1\right) \times 100 \approx 0.89\%$$

The final answer is $\boxed{0.89}$.

**Ground-Truth Answer**: 46.4157

**Model Output**: 0.89

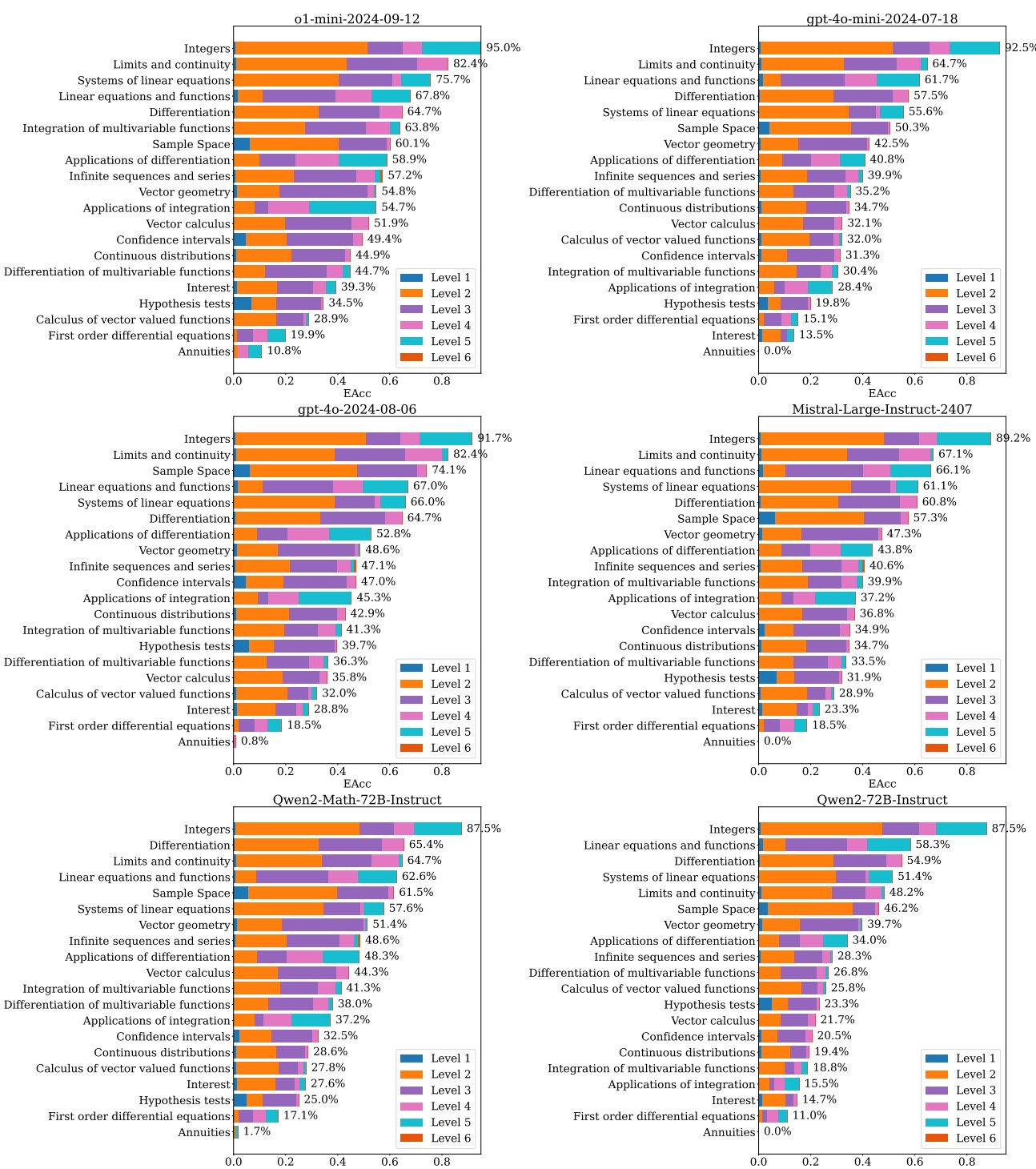

Figure 10: Model accuracy across topics

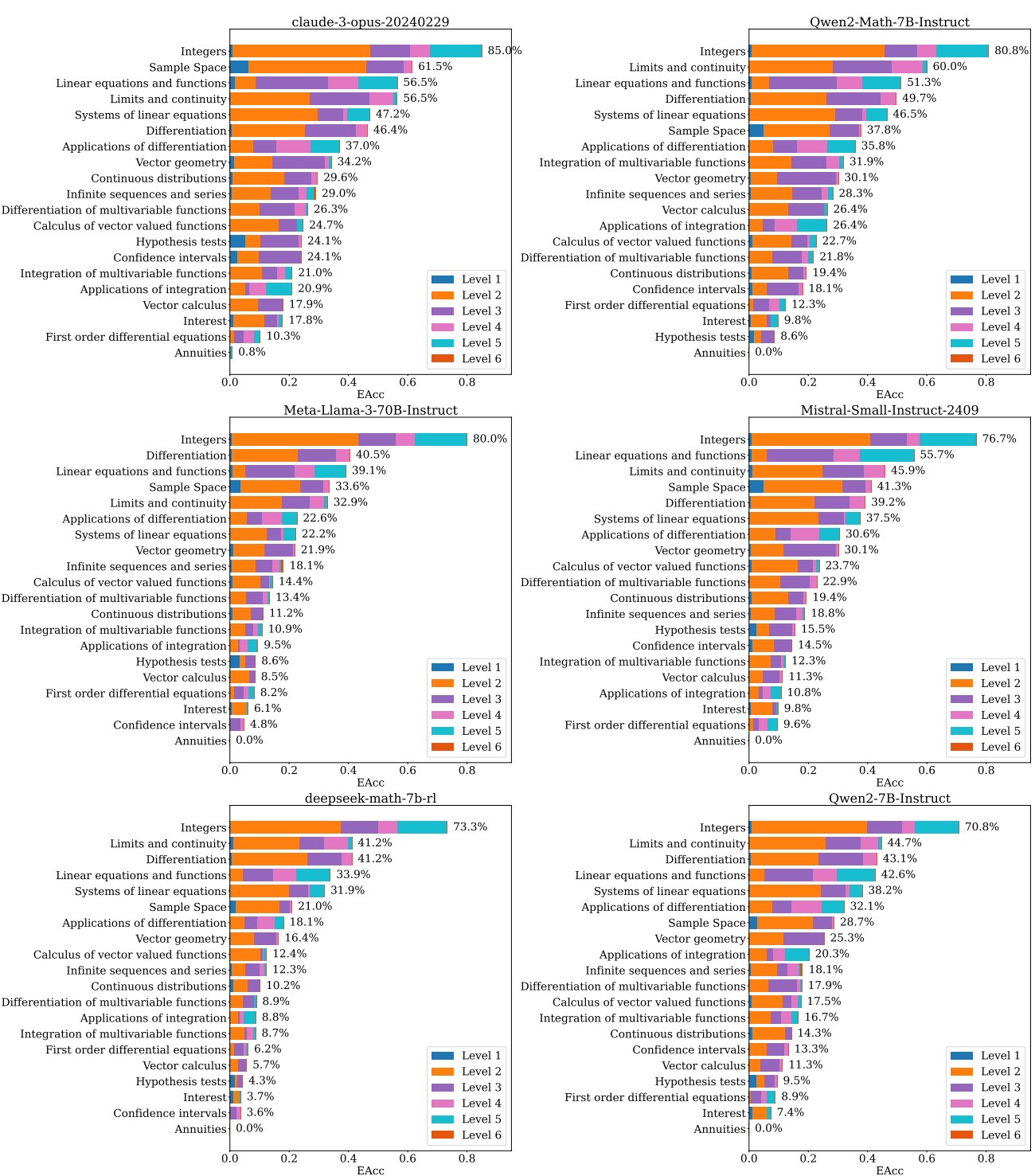

Figure 11: Model accuracy across topics

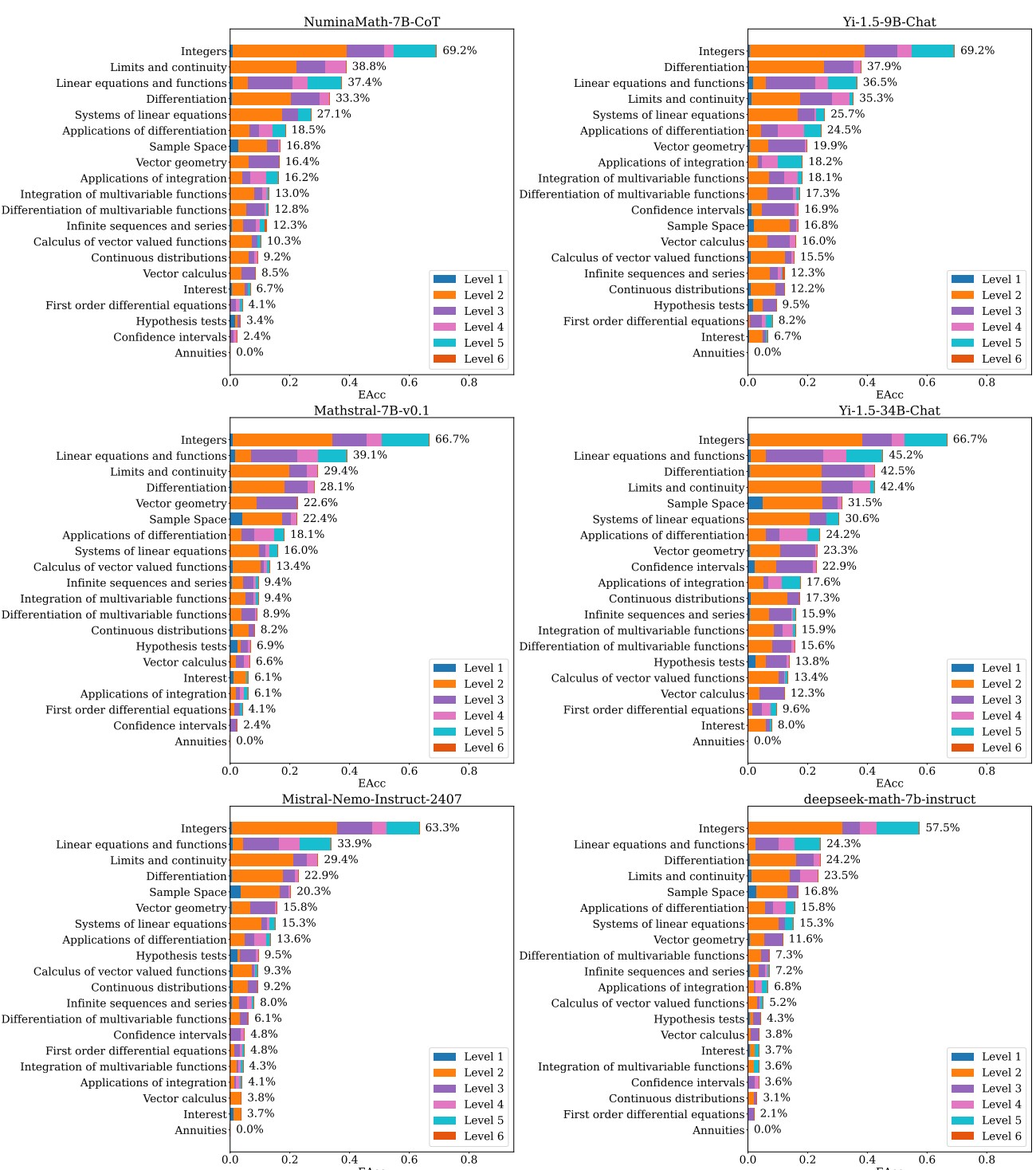

Figure 12: Model accuracy across topics

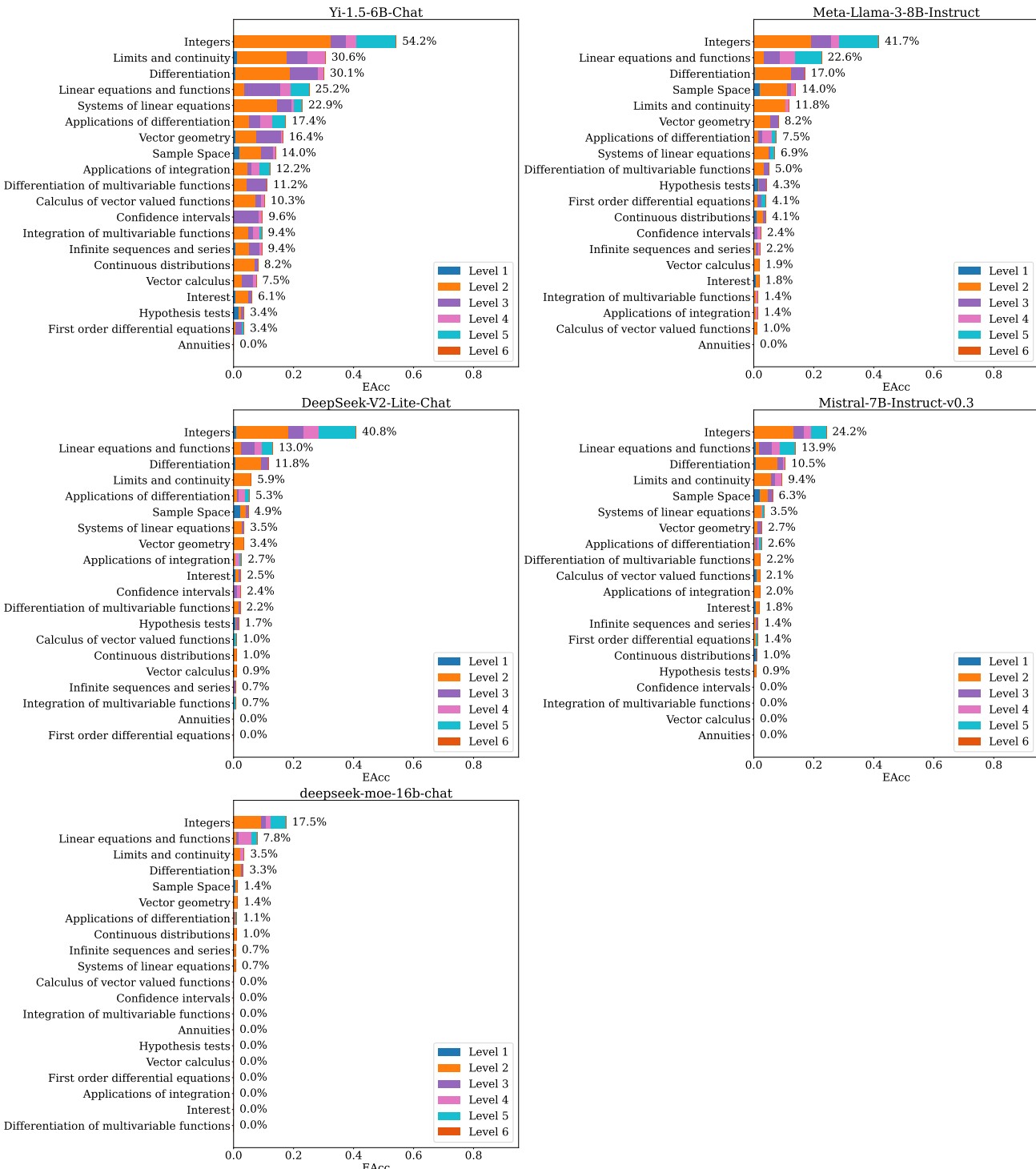

Figure 13: Model accuracy across topics

Table 32: **Results of PHP on UGMathBench** (all figures are in %). "diff." refers to the improvement over results in Table 4. The best results within column with "diff." are **bolded** and the worst results are underlined.

| Models | AAcc | diff. AAcc | EAcc | diff. EAcc | Δ | diff. Δ |
|---|---|---|---|---|---|---|
| *Closed-source LLMs* | | | | | | |
| GPT-4o-2024-08-06 | 60.73 | +0.36 | 50.44 | +0.50 | 10.29 | -0.14 |
| *Open-source Chat LLMs* | | | | | | |
| Yi-1.5-6B-Chat | 26.11 | -0.14 | 15.57 | +0.24 | 10.54 | -0.38 |
| Mistral-7B-Instruct | 10.76 | +0.20 | 4.41 | -0.03 | 6.35 | +0.23 |
| Qwen2-7B-Instruct | 36.38 | 0.0 | 25.29 | 0.14 | 11.09 | -0.14 |
| LLaMA3-8B-Instruct | 16.41 | -0.14 | 8.61 | -0.3 | 7.8 | +0.16 |
| Yi-1.5-9B-Chat | 34.50 | +0.21 | 21.75 | +0.63 | 12.75 | -0.42 |
| Mistral-Nemo-Instruct-2407 | 25.46 | +0.38 | 15.19 | -0.24 | 10.27 | +0.62 |
| DeepSeek-MOE-16B-Chat | 6.34 | +0.54 | 2.07 | +0.11 | 4.27 | +0.44 |
| DeepSeek-V2-Lite-Chat | 13.98 | +0.9 | 6.44 | +0.75 | 7.54 | +0.25 |
| Mistral-Small-Instruct-2409 | 39.66 | -0.56 | 27.90 | -0.94 | 11.76 | +0.38 |
| Yi-1.5-34B-Chat | 37.57 | +0.04 | 23.83 | -0.51 | 13.74 | +0.47 |
| LLaMA3-70B-Instruct | 33.93 | +0.31 | 23.24 | -0.03 | 10.69 | +0.34 |
| Qwen2-72B-Instruct | 47.89 | +0.13 | 36.08 | +0.30 | 11.81 | -0.17 |
| Mistral-Large-Instruct-2407 | 55.97 | +0.04 | 45.17 | +0.13 | 10.8 | -0.09 |
| *Specialized Mathematical LLMs* | | | | | | |
| DeepSeek-Math-7B-Instruct | 24.57 | +0.68 | 14.90 | +1.29 | 9.67 | **-0.61** |
| DeepSeek-Math-7B-RL | 29.42 | -0.23 | 19.58 | +0.24 | 9.84 | -0.47 |
| NuminaMath-7B-CoT | 27.81 | -1.99 | 16.62 | -2.19 | 11.19 | +0.20 |
| Mathstral-7B-v0.1 | 29.75 | **+1.24** | 19.26 | **+1.32** | 10.49 | -0.08 |
| Qwen2-Math-7B-Instruct | 42.65 | -1.08 | 31.18 | -1.08 | 11.47 | 0.0 |
| Qwen2-Math-72B-Instruct | 57.35 | + 0.32 | 46.04 | +0.19 | 11.31 | +0.13 |

Table 33: **Results across different subjects of PHP for GPT-4o**. (all figures are in %). "diff." refers to the improvement over results in Table 4. The best results within column with "diff." are **bolded** and the worst results are underlined.

| Subject | AAcc | diff. AAcc | EAcc | diff. EAcc | Δ | diff. Δ |
|---|---|---|---|---|---|---|
| Arithmetic | 92.11 | +0.29 | 88.47 | +1.20 | 3.64 | -0.91 |
| Algebra | 72.33 | +0.97 | 65.35 | +0.68 | 6.98 | + 0.29 |
| Set theory and logic | 71.98 | -1.45 | 57.97 | -1.45 | 14.01 | 0.0 |
| Trigonometry | 67.79 | -1.69 | 53.37 | +1.68 | 14.42 | -3.37 |
| Combinatorics | 78.79 | +1.14 | 65.91 | +4.55 | 12.88 | **-3.41** |
| Geometry | 67.49 | +3.31 | 54.04 | +2.49 | 13.45 | +1.82 |
| Calculus single-variable | 65.11 | +0.24 | 53.26 | +0.21 | 11.85 | +0.03 |
| Calculus multivariable | 49.85 | +0.21 | 38.53 | +0.30 | 11.32 | -0.09 |
| Linear Algebra | 62.85 | +0.47 | 52.81 | +1.20 | 10.04 | -0.73 |
| Number Theory | 65.22 | +1.45 | 54.35 | +2.18 | 10.87 | -0.73 |
| Financial Mathematics | 29.87 | +0.49 | 20.81 | +2.60 | 9.06 | -2.11 |
| Probability | 65.38 | -0.59 | 52.68 | -2.08 | 12.7 | +1.49 |
| Statistics | 60.27 | -0.16 | 48.88 | -0.50 | 11.39 | +0.34 |
| Complex Analysis | 70.59 | +1.31 | 58.82 | 0.0 | 11.77 | +1.31 |
| Differential Equations | 25.03 | 0.33 | 18.03 | -1.31 | 7.00 | +1.64 |
| Abstract Algebra | 53.17 | **+4.76** | 35.71 | **+7.14** | 7.46 | -2.38 |

