# OpenReview forum: "UGMathBench: A Diverse and Dynamic Benchmark for Undergraduate-Level Mathematical Reasoning with Large Language Models"
_ICLR.cc/2025/Conference — ICLR 2025 Poster_

### Official Review · Reviewer_LM6m · 2024-10-29

**Soundness:** 2
**Presentation:** 3
**Contribution:** 3
**Rating:** 6
**Confidence:** 4

**Summary:**

This paper introduces UGMathBench, a new benchmark for evaluating mathematical reasoning capabilities in LLMs at the undergraduate level. The benchmark addresses key limitations of existing evaluation methods by incorporating 5,062 diverse problems spanning 16 subjects, with each problem featuring three randomized versions to help detect potential test set contamination. The authors contribute two new metrics: effective accuracy (EAcc), which measures consistent problem-solving ability across variations, and reasoning gap ($\Delta$), which quantifies reasoning robustness by comparing average accuracy to EAcc. Detailed error analysis showing calculation errors as the primary failure mode and revealing inconsistencies in how models handle different versions of the same problem.

**Strengths:**

The functional variation approach represents a significant advancement over traditional math benchmarks. By systematically generating variations, it provides a robust mechanism to detect and account for potential test set contamination and overfitting - a critical concern given the widespread training of models on public mathematics content.

Additionally, the benchmark's difficulty is carefully calibrated to the current capabilities of LLMs while maintaining sufficient headroom for future improvements. This thoughtful positioning ensures the benchmark will remain relevant and discriminative as model capabilities advance, providing a stable metric for tracking progress in mathematical reasoning.

The authors offer valuable insights through: granular performance breakdowns across difficulty levels, subject-specific analysis that illuminates relative strengths and weaknesses, detailed examination of error modes that reveals systematic failure patterns (about half of the errors were due to calculation errors). This analysis not only establishes current baseline performance but also provides insights for model improvement and understanding of LLM mathematical capabilities.

**Weaknesses:**

The observed consistency in the reasoning gap metric (converging to approximately 10 across increasing ACC values) raises important methodological concerns. This finding contradicts established results from previous papers on functional variations and test set contamination, which demonstrated significant performance disparities among models with comparable ACC levels (a pattern typically indicative of overfitting). Two issues warrant discussion:

1. The apparent "reasoning gap" may primarily reflect sampling temperature effects rather than true reasoning capabilities. By definition, expected accuracy (EAcc) will be lower than actual accuracy (AAcc) due to the stochastic nature of the sampling process. This suggests the metric might be predominantly measuring a model's robustness to temperature variation rather than its reasoning capabilities.
2. Models previously demonstrated to overfit on the test set would be expected to exhibit substantially larger reasoning gaps. The absence of this pattern in the current results suggests that either: a) the metric is not effectively capturing reasoning differences, or b) the relationship between overfitting and reasoning capabilities requires re-examination. The authors claim that "These metrics help mitigate the impact of potential test set contamination," but their is no provided evidence that the proposed metrics detect contamination.

These observations suggest the need for additional validation of the metric's ability to meaningfully differentiate between genuine reasoning gaps and artifacts of the sampling process.

Minor:
Figure 1's layout appears overcrowded with difficult-to-read text sizes and fonts. Additionally, I would strongly reccomend against using pie charts as humans are bad at comparing areas.

**Questions:**

- What was the threshold used for deduplication and how many questions were filtered out?
- How well do students perform on the benchmark? (Assumming this data is availble/not private)
- Is it possible to derive the expected reasoning gap as a function of sampling temperature?
- How do models specifically overfit to one variation effect the reasoning gap?
- Were the "bad questions" found in the error analysis filtered out of the benchmark?

---

> ### Author Response · Authors · 2024-11-17
> **Response to Reviewer LM6m (1/2)**
>
> Dear Reviewer LM6m,
>
> We hope this comment finds you well! Thank you for your comments! We will address your concerns one by one as follows:
>
> > The apparent "reasoning gap" may primarily reflect sampling temperature effects rather than true reasoning capabilities. By definition, expected accuracy (EAcc) will be lower than actual accuracy (AAcc) due to the stochastic nature of the sampling process. This suggests the metric might be predominantly measuring a model's robustness to temperature variation rather than its reasoning capabilities.
>
> **There might be some misunderstandings**. As mentioned in Section 4.1 "Evaluation Settings", we "employ a greedy decoding strategy with temperature 0". There is no sampling temperature effects. As mentioned in Section 3.3, EAcc refers to "Effective Accuracy", not expected accuracy (EAcc). EAcc measures the fraction of test cases correctly solved across all versions. AAcc refers to "Average Accuracy", not "actual accuracy".
>
> > Models previously demonstrated to overfit on the test set would be expected to exhibit substantially larger reasoning gaps. The absence of this pattern in the current results suggests that either: a) the metric is not effectively capturing reasoning differences, or b) the relationship between overfitting and reasoning capabilities requires re-examination. The authors claim that "These metrics help mitigate the impact of potential test set contamination," but their is no provided evidence that the proposed metrics detect contamination.
>
> As suggested by Reviewer Ywbr, we have added Appendix D to show how different LLMs perform across 16 subjects. The results are shown in Table 13 - 28 in our revised manuscript and some discussion is given in Appendix D. From the results, the variation in the reasoning gap differs significantly across subjects.
>
> **We have also conducted experiments to explore how models specifically overfitting to a particular variation affect the reasoning gap in Section 5.5 in our revised manuscript**. We find that: as the proportion of the test set included in the training data increases, the reasoning gap also becomes more pronounced.
>
> > What was the threshold used for deduplication and how many questions were filtered out?
>
> Thank you for your question! We adopted subject-agnostic thresholds as follows:
>
> |                                          | **Threshold** | **Before** | **After** | **# Filter Out** |
> |------------------------------------------|---------------|------------|-----------|------------------|
> | Arithmetic                               | 0.92          | 845        | 322       | 523              |
> | Algebra                                  | 0.86          | 4663       | 583       | 4080             |
> | Set theory and logic                     | 0.96          | 94         | 69        | 25               |
> | Trigonometry                             | 0.91          | 399        | 178       | 221              |
> | Combinatorics                            | 0.89          | 140        | 88        | 52               |
> | Geometry                                 | 0.92          | 267        | 161       | 106              |
> | Calculus single-variable                 | 0.90          | 3234       | 982       | 2252             |
> | Calculus multivariable                   | 0.94          | 866        | 654       | 212              |
> | Linear Algebra                           | 0.93          | 1235       | 498       | 737              |
> | Number Theory                            | 0.92          | 68         | 46        | 22               |
> | Financial Mathematics                    | 0.92          | 653        | 346       | 307              |
> | Probability                              | 0.91          | 453        | 336       | 117              |
> | Statistics                               | 0.92          | 599        | 401       | 198              |
> | Complex Analysis                         | 0.95          | 129        | 51        | 78               |
> | Differential Equations                   | 0.92          | 741        | 305       | 436              |
> | Abstract Algebra                         | 0.94          | 58         | 42        | 16               |
> | **Grand Total**                          | -             | 14444      | 5062      | 9382             |
>
> **We have updated Section 3.2 "Data Cleaning and Deduplication" and Appendix B.2 (Table 10) accordingly in our revised manuscript.**
>
> > How well do students perform on the benchmark? (Assumming this data is availble/not private)
>
> We randomly select 100 problems to examine student performance using our grading system’ records, with each problem
> being completed by varying numbers of students ranging from 99 to 1,537. The average accuracy on the first attempt is 56.5%, while the average accuracy on the final attempt increased to 96.1%. **We have updated this part in Section 3.1 in our revised manuscript**.

---

> ### Author Response · Authors · 2024-11-17
> **Response to Reviewer LM6m (2/2)**
>
> > Is it possible to derive the expected reasoning gap as a function of sampling temperature?
>
> **There might be some misunderstandings**. As mentioned in Section 4.1 "Evaluation Settings", we "employ a greedy decoding strategy with temperature 0". There is no sampling temperature effects.
>
> > How do models specifically overfit to one variation effect the reasoning gap?
>
> **We have conducted experiments to explore how models specifically overfitting to a particular variation affect the reasoning gap in Section 5.5 in our revised manuscript**. We find that: as the proportion of the test set included in the training data increases, the reasoning gap also becomes more pronounced.
>
> > Were the "bad questions" found in the error analysis filtered out of the benchmark?
>
> At this moment, no. In our sample of 300 versions, 19 were identified as "bad problems," giving us an estimated occurrence of approximately 2.7% in our {\benchmark}. **We have added such details in Section 5.4 in our revised manuscript**. These problems do not impact our main claims, as no LLMs are able to solve these "bad questions. Additionally, these questions are actually in good quality (as students are able to solve them), simply not suitable for current LLMs to solve.
>
> > Minor: Figure 1's layout appears overcrowded with difficult-to-read text sizes and fonts. Additionally, I would strongly reccomend against using pie charts as humans are bad at comparing areas.
>
> Thank you for pointing this out! We have changed the pie charts to bar charts and changed the fonts in our revised manuscript.
>
> We hope our changes have addressed your concerns. Feel free to further discuss with us at this stage!
>
> Best Regards,
>
> Authors

---

> ### Author Response · Authors · 2024-11-23
>
> Dear Reviewer LM6m,
>
> Thank you again for your feedback and comments!
>
> We hope our changes have addressed your concerns. If you have any further questions, please do not hesitate to let us know. We are ready to have further discussion with you at this stage!
>
> Best,
>
> Authors

---

> > ### Comment · Reviewer_LM6m · 2024-11-25
> >
> > Thank you for the follow up and addressing my misunderstanding with the sampling temperature. I'm still a little hesitant about the benifits of using the new proposed metrics over standard accuracy as there is a very consistent relationship between them and I'm not entirely sure what new information we get compared to standard accuracy measures. Perhaps a more in depth and expansive investigation into contamination would prove this a more useful metric. However, I don't see anny other issues with the soundness of the metric and I have decided to increase my score accordingly.

---

> > > ### Author Response · Authors · 2024-11-25
> > >
> > > Thank you for your constructive feedback, and for taking the time to engage with our work. We appreciate your response and are glad to have clarified your earlier concerns.
> > >
> > > For your concerns about our proposed metrics, we further explain as follows:
> > >
> > > - We are not suggesting these metrics as replacements of the standard accuracy but rather as complementary tools to provide a more comprehensive evaluation of model performance. Specifically, we combine standard accuracy with our proposed metrics (EAcc and reasoning gap) to offer a fuller picture of model performance and robustness.
> > >
> > > -  From the calculation of EAcc and reasoning gap in Section 3.3, EAcc can measure how consistently a model performs the same task under small variations in input values. Reasoning gap quantifies the robustness of a model’s reasoning process, allowing us to better understand its ability to handle subtle variable changes or perturbations reliably.
> > >
> > > - When evaluating a single version of a model, EAcc is identical to standard accuracy. However, in cases where overfitting to a specific test set occurs, accuracy alone may misleadingly suggest an inflated level of model performance. As the number of versions (or variations in datasets) increases, EAcc starts to differentiate itself from standard accuracy, capturing aspects of performance that standard metrics overlook.
> > >
> > > - We can also view it from another perspective: When more versions of our test set are released in the future, EAcc and reasoning gap will offer a more convenient and nuanced way to assess overall performance trends.

---

> ### Comment · Reviewer_LM6m · 2024-11-27
>
> Thank you for your response. However, with the currently provided results, I do not see convincing evidence that EAcc measures something particularly novel or insightful. There are already methods to detect test set contamination (For instance simply looking at the drop in accuracy when testing a model on variation datasets) and it is unclear what advantage this metric has over this.
>
> With that said, I do think this is a very solid dataset and having the ability to generate question variations is very useful, thus I will leave my score as is.

---

### Official Review · Reviewer_2zWK · 2024-10-30

**Soundness:** 3
**Presentation:** 3
**Contribution:** 3
**Rating:** 8
**Confidence:** 4

**Summary:**

Authors propose a new benchmark namely UGMathBench comprising of undergraduate level math problems for testing mathematical reasoning abilities of LLMs. Authors also propose two metrics namely effective accuracy and rasoning gap and perform extensive experiments with multiple open and closed source LLMs showing that there is still a big scope for improving mathematical reasoning abilities of LLMs.

**Strengths:**

1) Dataset is comprehensive consisting of 5062 problems including problems from different areas of mathematics.

2) I acknowledge the efforts made by the authors in cleaning the data by performing deduplication ensuring that no problems are repeated and including multiple versions of the problems for thorough evaluation of mathematical reasoning abilities of LLMs.

3) Authors have promised to make the benchmark dynamic as in to include new and more tougher problems in the future

4) The proposed metrics namely effective accuracy and reasoning gap provide an effective way to evaluate the reasoning abilities of LLMs.

**Weaknesses:**

1) Authors should experiment with more advanced prompting techniques like progressive hint prompting [1]. Progressive hint prompting allows LLMs to use previously generated answer as a hint which allows LLMs to correct/ refine their mistakes. I believe that this experiment would provide crucial insights if LLMs can solve tough mathematical problems with refinement. I will increase my score if authors perform this experiment.

2) Authors mention in line 477 that there are some number of "bad questions" present in the benchmark which may not be suitable for LLMs to solve. Pls mention the number and percentage of bad questions present in the benchmark as it directly effects the quality of the benchmark.

**Questions:**

what are the limitations of the current work? Pls include a description of limitiations of current work too as it is important

---

> ### Author Response · Authors · 2024-11-17
> **Response to Reviewer 2zWK**
>
> Dear Reviewer 2zWK,
>
> We hope this comment finds you well! Thank you for your comments! We will address your concerns one by one as follows:
>
> > Authors should experiment with more advanced prompting techniques like progressive hint prompting [1]. Progressive hint prompting allows LLMs to use previously generated answer as a hint which allows LLMs to correct/ refine their mistakes. I believe that this experiment would provide crucial insights if LLMs can solve tough mathematical problems with refinement. I will increase my score if authors perform this experiment.
>
> Thank you for your helpful suggestion! **We have added PHP as our experiments in Section 5.5 in our revised manuscript. The detailed results are given in Appendix G (Table 32 and 33)**. We find that:
>
> -  Although PHP improves EAcc and AAcc for most LLMs, the enhancements are not significant, indicating considerable room for
> future development.
>
> - The impact of refinement for GPT-4o varies across different subjects. For instance, PHP improves GPT-4o’s performance in
> abstract algebra by 7.14\%, yet reduces its performance in probability by 2.08\% in terms of EAcc.
>
> > Authors mention in line 477 that there are some number of "bad questions" present in the benchmark which may not be suitable for LLMs to solve. Pls mention the number and percentage of bad questions present in the benchmark as it directly effects the quality of the benchmark.
>
> Thank you for pointing this out. In our sample of 300 versions, 19 were identified as "bad problems," giving us an estimated occurrence of approximately 2.7\% in our {\benchmark}. **We have added such details in Section 5.4 in our revised manuscript**. These problems do not impact our main claims, as no LLMs are able to solve these "bad questions. Additionally, these questions are actually in good quality (as students are able to solve them), simply not suitable for current LLMs to solve.
>
> > what are the limitations of the current work? Pls include a description of limitiations of current work too as it is important
>
> Thank you for your comments! **We have added a "Limitation" section in Appendix to discuss the limitations of our work in our revised manuscript**:
>
> "This work has several limitations. First, UGMathBench focuses on text-only reasoning, whereas some undergraduate-level math problems require images for their solutions. Developing a multimodal benchmark for undergraduate-level mathematics will be future work. Second, UGMathBench is designed as an English-language benchmark. Extending UGMathBench to support multiple languages could be an interesting avenue for future research. Third, the number of problems in certain subjects is limited. Expanding these subjects would be valuable."
>
> We hope our changes have addressed your concerns. Feel free to further discuss with us at this stage!
>
> Best Regards,
>
> Authors

---

> > ### Comment · Reviewer_2zWK · 2024-11-17
> >
> > Thank you for addressing all my questions and going the extra mile to conduct the PHP experiment. The result of PHP experiments are promissing showing future scope. I believe the current benchmark represents a significant contribution for evaluating the mathematical reasoning capabilities of LLMs. The paper demonstrates a high level of rigor in the experiments conducted and maintains an overall strong quality. The limitations section is particularly noteworthy, as it provides a clear perspective on future directions and paints an exciting picture of the possibilities ahead. I sincerely hope the authors pursue the directions outlined in the limitations section.
> >
> > In light of the rebuttals, I have revised my score to reflect my appreciation for the work and would strongly advocate for the paper's acceptance!

---

> > > ### Author Response · Authors · 2024-11-17
> > > **Response to Reviewer 2zWK**
> > >
> > > Thank you for your prompt response, really amazing! Thank you very much for your thoughtful and encouraging feedback on our paper!
> > >
> > > The suggestion to incorporate PHP is insightful and interesting! We are particularly grateful for your commendation of the limitations section! Your thoughtful remarks and support play a vital role in advancing our research, and we are grateful to have such an engaged and supportive reviewer!
> > >
> > >
> > > Thank you once again for your invaluable feedback and for contributing to enhancing the quality of our work. We are looking forward to building upon this research and exploring the exciting possibilities ahead.

---

### Official Review · Reviewer_ZpsQ · 2024-11-01

**Soundness:** 4
**Presentation:** 4
**Contribution:** 3
**Rating:** 6
**Confidence:** 5

**Summary:**

This work underscores the limitations of current mathematical benchmarks due to the rapid advancements in mathematical reasoning capabilities of LLMs and issues of test-set contamination. To address these challenges, the authors introduce UGMathBench, a comprehensive benchmark designed to effectively evaluate these capabilities. UGMathBench consists of over 5,000 problems across various subjects, aiming to provide a fair assessment of LLM performance. It addresses common issues in existing benchmarks by offering a dynamic dataset that can generate different question variations by setting random seeds. Additionally, the authors propose new metrics to evaluate accuracy and reasoning robustness, facilitating a deeper analysis of the challenges LLMs face in mathematical reasoning.

**Strengths:**

1. **Dynamic Dataset:** UGMathBench provides multiple randomized versions of each problem, allowing for a robust assessment of LLM reasoning abilities. This approach enables a more rigorous evaluation by examining how well models handle variations of the same problem.

2. **Diversity:** The benchmark covers a wide range of mathematical topics, ensuring comprehensive subject matter coverage.

3. **Difficulty Level:** UGMathBench includes undergraduate-level problems, challenging LLMs with more advanced content.

4. **Innovative Evaluation Metrics:** The paper introduces metrics like Effective Accuracy (EAcc) and Reasoning Gap (∆) to measure LLM performance. EAcc captures the percentage of problems correctly solved across all problem variations, while the Reasoning Gap quantifies the robustness of reasoning when problems are modified slightly.

5. **High-Quality Problem Collection:** The problems are carefully curated, cleaned, and formatted from an online undergraduate-level homework grading system, ensuring quality and relevance.

6. **Fully Public Benchmark:** UGMathBench is publicly available, enabling LLM developers and users to conduct thorough error analysis.

7. **Detailed Error Analysis:** The benchmark provides insights into the current limitations of LLMs in mathematical reasoning, guiding future directions for improvement.

**Weaknesses:**

1. **Premise of Problem Variants:** The paper assumes that "a model capable of solving a problem through reasoning should also be able to solve all its variants under variable disturbance." The benchmark achieves this by randomizing numbers in each question to create multiple versions while theoretically maintaining the problem’s difficulty and intrinsic properties. However, simply changing certain values can fundamentally alter the problem. For example, modifying the denominator in $\int_{a}^{b} \frac{x^3}{x^4 - 1} \, dx$ to $\int_{a}^{b} \frac{x^3}{x^4 + 1} \, dx$ requires a completely different solution approach. Similarly, changing the series $\sum_{n=1}^{\infty} \frac{1}{n^2}$, which converges to $\frac{\pi^2}{6}$, to $\sum_{n=1}^{\infty} \frac{1}{n}$ transforms it into the divergent harmonic series. These examples show that certain modifications in mathematical problems lead to essientially different problems, which may not align with the benchmark’s intended assessment of consistent reasoning across equivalent variants.

2. **Precision in Numerical Calculations:** Classifying the example in **Table 12** as a "numerical calculation error" might be overly strict, given that the minor difference between 12305.72 and 12305.670999 likely falls within acceptable rounding tolerance. Unless the benchmark is meant to test extreme precision, such small discrepancies shouldn’t necessarily indicate an error. Accuracy assessments could allow for minor rounding differences or specify a standard precision level (e.g., significant figures or decimal places) for intermediate calculations to ensure fair evaluation.

**Questions:**

1. How do you ensure that changes to specific values within a problem do not fundamentally alter its nature or required solution approach? Given that small modifications can impact mathematical properties such as convergence and factorization, are there any safeguards or verification steps to confirm that randomized problem variants retain similar problem structures, difficulty levels, and solution approaches as the original?
2. Would it be feasible to incorporate a tolerance level for minor discrepancies in numerical answers, especially those influenced by floating-point precision errors? For instance, a defined rounding tolerance (e.g., within 0.01%) could help accommodate acceptable computational variations and more accurately reflect the model’s reasoning accuracy.

---

> ### Author Response · Authors · 2024-11-17
> **Response to Reviewer ZpsQ**
>
> Dear Reviewer ZpsQ,
>
> We hope this comment finds you well! Thank you for your comments! We will address your concerns one by one as follows:
>
> > How do you ensure that changes to specific values within a problem do not fundamentally alter its nature or required solution approach? Given that small modifications can impact mathematical properties such as convergence and factorization, are there any safeguards or verification steps to confirm that randomized problem variants retain similar problem structures, difficulty levels, and solution approaches as the original?
>
> Thank you for your question! In fact, the problems in our grading system are generated by programs, with specific range for variables to guarantee the required solution approach and correctness (As implicitly mentioned in Appendix B.1 Line 1024 in the original manuscript).  **We have made it more clear and added two examples in Appendix A.1 and B.1 (Figure 7 and 8, Listing 1) to show this point**. For your convenience, we briefly discuss these examples here:
>
> - For question $\int x^5 cos(x^3) dx$, when we generate this version of problem, the following ranges of variables are considered: \$a = random(2, 10); \$ expnt = -1 + 2* \$ a;. This meticulous setup maintains the consistency of the concepts, techniques, and solutions
> involved in different versions of each problem.
>
> - For question to determine the convergence of series $\sum_1^\infty \frac{sin (n)}{n^{6.5}}$, when we generate this version of problem, the following range of variable is considered: $a = random(2,10,0.1). This setup makes sure this series is convergent.
>
> > Precision in Numerical Calculations: Classifying the example in Table 12 as a "numerical calculation error" might be overly strict, given that the minor difference between 12305.72 and 12305.670999 likely falls within acceptable rounding tolerance. Unless the benchmark is meant to test extreme precision, such small discrepancies shouldn’t necessarily indicate an error. Accuracy assessments could allow for minor rounding differences or specify a standard precision level (e.g., significant figures or decimal places) for intermediate calculations to ensure fair evaluation.
>
> Thank you for pointing this out! We apologize for the wrong example provided. The example in Table 12 also surprises us when we see your Comments: we set the relative precision error to be 1e-3 in our evaluation. After checking, we find that one of the authors picked a wrong example whose answer was judged to be correct. We were rushing to the deadline and did not discover this error. Now we have updated the example in Table 30 in our revised manuscript.
>
>
> > Would it be feasible to incorporate a tolerance level for minor discrepancies in numerical answers, especially those influenced by floating-point precision errors? For instance, a defined rounding tolerance (e.g., within 0.01%) could help accommodate acceptable computational variations and more accurately reflect the model’s reasoning accuracy.
>
> Yes. We **have set the relative tolerance level to 1e-3 in our evaluation**, which is quite looser than the criterion for OlympaidBench [1] (1e-5) and OlympicArena [2] (1e-8). We have also added a figure (Figure 9) to illustrate the relative numerical error for OpenAI-o1-mini in Appendix E. We can see from the figure that 1e-3 is a suitable choice.
>
> We acknowledge the potential issue of numerical calculation errors in LLMs. We believe that integrating tool-supported reasoning could be a promising approach to mitigate this issue in the future [3, 4].
>
>
>
> We hope our changes have addressed your concerns. Feel free to further discuss with us at this stage!
>
> Best Regards,
>
> Authors
>
> [1] He, C., Luo, R., Bai, Y., Hu, S., Thai, Z. L., Shen, J., ... & Sun, M. (2024). Olympiadbench: A challenging benchmark for promoting agi with olympiad-level bilingual multimodal scientific problems. arXiv preprint arXiv:2402.14008.
>
> [2] Huang, Z., Wang, Z., Xia, S., Li, X., Zou, H., Xu, R., ... & Liu, P. (2024). OlympicArena: Benchmarking Multi-discipline Cognitive Reasoning for Superintelligent AI. arXiv preprint arXiv:2406.12753.
>
> [3] Gou, Z., Shao, Z., Gong, Y., Shen, Y., Yang, Y., Huang, M., ... & Chen, W. (2023). Tora: A tool-integrated reasoning agent for mathematical problem solving. arXiv preprint arXiv:2309.17452.
>
> [4] Yang, A., Zhang, B., Hui, B., Gao, B., Yu, B., Li, C., ... & Zhang, Z. (2024). Qwen2. 5-math technical report: Toward mathematical expert model via self-improvement. arXiv preprint arXiv:2409.12122.

---

> > ### Comment · Reviewer_ZpsQ · 2024-11-17
> >
> > Thank you for addressing my questions thoroughly. Your responses clarified my concerns, and I will be raising my rating score accordingly.

---

> > > ### Author Response · Authors · 2024-11-17
> > >
> > > Thank you for your quite reply and for taking the time to review our work! We are delighted that our responses addressed your questions! Thank you for your helpful comments to improve the quality of our manuscript!

---

### Official Review · Reviewer_Ywbr · 2024-11-04

**Soundness:** 4
**Presentation:** 4
**Contribution:** 3
**Rating:** 6
**Confidence:** 4

**Summary:**

The paper introduces UGMathBench, an undergraduate-level math reasoning benchmark for LLMs. THe benchmark includes 5,062 problems across 16 subjects, with randomized variations to prevent test-set contamination and evaluate reasoning consistency. All problems are taken from the authors’ school curricula. The authors also propose a variety of metrics that could theoretically be applied to other benchmarks (with additional work) to enable more robust benchmarking of LLM abilities. Testing 23 LLMs, the highest EAcc achieved was 56.3%, highlighting challenges for current models.

**Strengths:**

* The authors create a novel undergraduate-level benchmark for math reasoning.
* The proposal of the Effective Accuracy, Average Accuracy, Reasoning Gap, and Robustness Efficiency metrics are interesting, and are hopefully adopted by math benchmarks as a whole to promote more robust benchmarking.
* The results on each of these metrics are described at length, with potential reasons explored for any interesting/unexpected results.
* UGBench is fine grained, with each question being labeled with a main topic and subtopic, enabling a more granular study of various LLMs’ strengths and weaknesses.
* The appendix is robust and includes detailed information about the creation and distribution of the benchmark.

**Weaknesses:**

* The overall discussion of results feels slightly generic (besides what is driven by the metrics proposed in the paper). It demonstrates a lot of what has already been established (namely, undergraduate math being a challenging task, closed source models performing better than open source, bigger model size = better performance, specialized LLMs’ improved performance over unspecialized ones, etc.) UGBench is fairly uniquely positioned in how granular it is (in terms of each question being associated with a topic/subtopic). I believe this could lead to interesting comparisons between models and which skill sets certain families of models succeed/fail on.

**Questions:**

* Same as weaknesses — I think there are possibly more interesting analyses that could be conducted given how rich the data in UGMathBench is (as opposed to the usual open vs. small, regular vs. specialized, etc.)

---

> ### Author Response · Authors · 2024-11-17
> **Response to Reviewer Ywbr**
>
> Dear Reviewer Ywbr,
>
> We hope this comment finds you well! Thank you for your comments! We will address your concerns one by one as follows:
>
> > UGBench is fairly uniquely positioned in how granular it is (in terms of each question being associated with a topic/subtopic). I believe this could lead to interesting comparisons between models and which skill sets certain families of models succeed/fail on.
>
> Thank you for your helpful suggestion! In the original manuscript, we have included discussion about performance across different subjects and difficulty levels in Section 5.3. **As suggested, we have made the following changes to incorporate discussions about performance across different subjects and topics about individual LLMs**:
>
> - We have added Appendix D to show how different LLMs perform across 16 subjects. The results are shown in Table 13 - 28 in our revised manuscript and some discussion is given in Appendix D.
>
> - We have added Appendix F to show how different LLMs perform across a subset of topics. The results are shown in Figure 10 - 13 in our revised manuscript. We find that different LLMs exhibit varying performance patterns across these topics, and even
> models within the same family show differences in their rankings.
>
> > I think there are possibly more interesting analyses that could be conducted given how rich the data in UGMathBench is (as opposed to the usual open vs. small, regular vs. specialized, etc.)
>
> Thank you for your helpful comments! **We have adjusted our manuscript to include more interesting analysis in Section 5.5 and Appendix G**:
>
> As suggested by Reviewer 2zWK, we have included an analysis about how prompting with refinement help improve the performance of different LLMs in Section 5.5 and Appendix G in our revised manuscript. From Table 33, we observe that **the effects of refinement are quite different across different subjects**.
>
> We hope our changes have addressed your concerns. Feel free to further discuss with us at this stage!
>
> Best Regards,
>
> Authors

---

> > ### Comment · Reviewer_Ywbr · 2024-11-25
> > **Thank you!**
> >
> > Thank you for making the changes, especially the data on granularity! These tables set the stage for interesting future research. I support the acceptance of this paper.

---

> > > ### Author Response · Authors · 2024-11-26
> > >
> > > Thank you for your response and positive feedback! We appreciate your suggestions to improve our manuscript's quality!

---

### Meta-Review · Area_Chair_jPp8 · 2024-12-21

**Metareview:**

a) Summary of Scientific Claims and Findings:
The paper introduces UGMathBench, a comprehensive benchmark for evaluating mathematical reasoning capabilities of Large Language Models (LLMs) at the undergraduate level. The benchmark contains 5,062 problems across 16 subjects and 111 topics, with each problem having three randomized versions. The authors propose two key metrics: Effective Accuracy (EAcc) and Reasoning Gap (Δ). Testing on 23 LLMs showed a highest EAcc of 56.3% by OpenAI-o1-mini, with significant reasoning gaps across models.

(b) Strengths:
- Comprehensive coverage of undergraduate math subjects with diverse problem types
- Dynamic dataset design with multiple randomized versions to prevent test-set contamination
- Novel evaluation metrics (EAcc and Reasoning Gap) for assessing reasoning robustness
- Detailed error analysis and granular performance breakdowns
- High-quality problem collection from actual undergraduate curricula
- Public availability enabling thorough error analysis

(c) Weaknesses:
- Some debate over the effectiveness of proposed metrics in detecting test set contamination
- Presence of "bad questions" (~2.7%) that current LLMs cannot solve
- Limited discussion of more advanced analysis possibilities given the rich dataset
- Some limitations in coverage (text-only problems, English-only, limited problems in certain subjects)

(d) Reasons for Accept:
The paper makes a significant contribution to the field by providing a robust, diverse benchmark for mathematical reasoning. The dynamic nature of the dataset and proposed evaluation metrics offer valuable tools for assessing LLM capabilities. While there are some limitations, the authors have addressed reviewer concerns and demonstrated the benchmark's utility through comprehensive experiments.

**Additional Comments On Reviewer Discussion:**

Key Points Raised and Addressed:
1. Reviewer Ywbr suggested more granular analysis of model performance:
- Authors added detailed performance analysis across 16 subjects (Appendix D)
- Added analysis across topics (Appendix F)

2. Reviewer ZpsQ raised concerns about problem variants and numerical precision:
- Authors clarified the variable range constraints ensuring consistent problem difficulty
- Explained the 1e-3 relative tolerance level for numerical answers
- Updated examples to better reflect evaluation criteria

3. Reviewer 2zWK requested progressive hint prompting experiments:
- Authors added PHP experiments showing varied improvements across subjects
- Added limitations section discussing current constraints

4. Reviewer LM6m questioned the metrics and methodology:
- Authors clarified greedy decoding strategy (temperature=0)
- Added evidence of metric effectiveness through subject-wise analysis
- Provided detailed deduplication thresholds and student performance data

All reviewers ultimately supported acceptance after their concerns were addressed through the rebuttal period. The authors were responsive and made substantial improvements to the manuscript based on reviewer feedback.

---

### Decision · Program_Chairs · 2025-01-22

Accept (Poster)